# KChIP3 coupled to Ca²⁺ oscillations exerts a tonic brake on baseline mucin release in the colon

Gerard Cantero-Recasens[1], Cristian M Butnaru[1], Miguel A Valverde[2], José R Naranjo[3,4], Nathalie Brouwers[1], Vivek Malhotra[1,5,6]*

[1]Centre for Genomic Regulation, The Barcelona Institute of Science and Technology, Barcelona, Spain; [2]Laboratory of Molecular Physiology and Channelopathies, Department of Experimental and Health Sciences, Pompeu Fabra University, Barcelona, Spain; [3]Spanish Network for Biomedical Research in Neurodegenerative Diseases, Madrid, Spain; [4]National Biotechnology Center, Madrid, Spain; [5]Universitat Pompeu Fabra, Barcelona, Spain; [6]Institució Catalana de Recerca i Estudis Avançats, Barcelona, Spain

**Abstract** Regulated mucin secretion from specialized goblet cells by exogenous agonist-dependent (stimulated) and -independent (baseline) manner is essential for the function of the epithelial lining. Over extended periods, baseline release of mucin can exceed quantities released by stimulated secretion, yet its regulation remains poorly characterized. We have discovered that ryanodine receptor-dependent intracellular Ca²⁺ oscillations effect the dissociation of the Ca²⁺-binding protein, KChIP3, encoded by *KCNIP3* gene, from mature mucin-filled secretory granules, allowing for their exocytosis. Increased Ca²⁺ oscillations, or depleting KChIP3, lead to mucin hypersecretion in a human differentiated colonic cell line, an effect reproduced in the colon of *Kcnip3⁻/⁻* mice. Conversely, overexpressing KChIP3 or abrogating its Ca²⁺-sensing ability, increases KChIP3 association with granules, and inhibits baseline secretion. KChIP3 therefore emerges as the high-affinity Ca²⁺ sensor that negatively regulates baseline mucin secretion. We suggest KChIP3 marks mature, primed mucin granules, and functions as a Ca²⁺ oscillation-dependent brake to control baseline secretion.

**Editorial note:** This article has been through an editorial process in which the authors decide how to respond to the issues raised during peer review. The Reviewing Editor's assessment is that all the issues have been addressed (see decision letter).

DOI: https://doi.org/10.7554/eLife.39729.001

*For correspondence:
vivek.malhotra@crg.eu

## Introduction

Mucins, encoded by 21 different genes in mammals, are the major components of the mucus layer, which provides the first line of defense against pathogens and allergens to the lining of the respiratory, urinary, gastrointestinal and reproductive tracts (*Boucher, 2007*; *Kesimer et al., 2013*; *Russo et al., 2006*). Specialized epithelial goblet cells secrete gel-forming mucins (MUC2, MUC5AC, MUC5B and MUC6) that control the rheological properties of the mucus layer (*Davis and Dickey, 2008*; *Kreda et al., 2012*; *Thornton et al., 2008*). Mucins are synthesized in the endoplasmic reticulum (ER) where they undergo core glycosylation and reach up to 500 kDa in apparent molecular weight. The fully assembled mucins are then exported to the Golgi complex where, after extensive glycosylation, they reach sizes of up to 2.5 million daltons (*Sheehan et al., 2004*; *Thornton et al., 2008*). These heavily glycosylated mucins are then packed into secretory granules that can be up to a micron in size and occupy almost 75% of the cytoplasmic volume (*Curran and Cohn, 2010*). The

granules mature to produce highly condensed mucins and finally a subset of granules fuse with the plasma membrane by SNARE-mediated fusion (*Adler et al., 2013*). In the extracellular space, condensed mucins undergo a change in their organization to a gel-like form to compose a layer of mucus that coats the extracellular surface of the epithelium (*Thornton et al., 2008*).

Mucin granule exocytosis is a $Ca^{2+}$-regulated process that can occur at basal level (Baseline Mucin Secretion, BMS) and by an exogenously supplied agonist dependent release (Stimulated Mucin Secretion, SMS) (*Adler et al., 2013*; *Rossi et al., 2004*). Importantly, baseline mucin secretion could be the preponderant mode of goblet cell mucin release in healthy tissue, and may contribute predominately to mucus formation in allergic and infectious inflammation (*Zhu et al., 2015*). The $Ca^{2+}$ binding properties of the well-characterized Synaptotagmin 2 (Syt2: low affinity and high cooperativity (*Xu et al., 2007*)) provide the high fidelity necessary for stimulated secretion. Syt2, however, does not have a role in baseline mucin secretion (*Tuvim et al., 2009*). What then is the identity of the corresponding high-affinity $Ca^{2+}$ sensor required for baseline (agonist-independent) mucin secretion?

Our genome-wide screen identified new proteins that regulate mucin secretion, such as TRPM5, a $Na^+$ channel that controls extracellular $Ca^{2+}$ entry into cells (*Mitrovic et al., 2013*). Also identified in the pool of hits from this screen, was a high-affinity $Ca^{2+}$-binding protein, KChIP3 (potassium voltage-gated channel interacting protein 3), also known as DREAM and Calsenilin (*An et al., 2000*; *Buxbaum et al., 1998*; *Carrión et al., 1999*), which is encoded by the gene *KCNIP3*. KChIP3 is a member of the neuronal $Ca^{2+}$ sensor protein (NCS) family that codes a 29 kDa multifunctional $Ca^{2+}$-binding protein (*Carrión et al., 1999*). It has three functional EF hands and can bind two $Ca^{2+}$ ions with high affinity and another with lower affinity (*Lusin et al., 2008*; *Osawa et al., 2005*). Curiously, the $Ca^{2+}$ regime under which KChIP3 senses $Ca^{2+}$, at < 1 µM, is an order of magnitude lower than the levels required to regulate stimulated secretion, which is estimated at 10 µM in the vicinity of the exocytic machinery in goblet cells (*Rossi et al., 2004*; *Yu et al., 2007*). We therefore reasoned that KChIP3 could be the key to controlling baseline mucin secretion ordinarily associated with lower intracellular $Ca^{2+}$ concentrations. In other words, while Syt2 functions in stimulated secretion as a low-affinity $Ca^{2+}$ sensor, KChIP3 could be the high-affinity $Ca^{2+}$ sensor for baseline secretion.

Here we demonstrate that KChIP3 is the $Ca^{2+}$-sensing brake that controls agonist-independent baseline mucin secretion in tissue culture colonic goblet cells and in the mouse colon.

## Results

### KChIP3 is required for baseline mucin secretion in colonic goblet cells

Previously, we showed that differentiation of HT29-18N2 cells into mucin-secreting cells results in upregulation of mucins and many of the genes that are required for its secretion (*Mitrovic et al., 2013*). We re-tested mRNA levels of KChIP3 in differentiated HT29-18N2 goblet cells compared to undifferentiated cells. The data reveal a 3.3-fold increase in the mRNA levels of KChIP3 (*Figure 1A*). Next, we generated a stable HT29-18N2 cell line depleted of KChIP3 (KChIP3-KD) and a stable HT29-18N2 cell line expressing KChIP3 tagged with GFP at the C-terminus (KChIP3-GFP). RNA was extracted from differentiated control, KChIP3-KD and KChIP3-GFP HT29-18N2 cells and the levels of KChIP family members monitored by qPCR. KChIP3-KD cells showed an 80% reduction in *KCNIP3* mRNA levels, while levels of the other KChIP family members were unaffected (*Figure 1—figure supplement 1A*). In addition, expression of KChIP3-GFP, which was confirmed by western blot (*Figure 1—figure supplement 1B*), did not significantly affect the levels of the other KChIP family members (*Figure 1—figure supplement 1C*). The commercial antibodies do not detect endogenous levels of KChIP3, therefore we can only provide a value of how much KChIP3 is overexpressed in KChIP3-GFP cell line compared to endogenous KChIP3 at the mRNA level. We used these cell lines to measure MUC5AC secretion in the absence (baseline) or presence (stimulated) of the physiological stimulus ATP (100 µM in a solution containing 1.2 mM $CaCl_2$). After 30 min at 37°C, extracellular medium was collected and dot blotted with anti-MUC5AC antibody as described previously (*Mitrovic et al., 2013*). Within 30 min, our results reveal a strong (2.5-fold) increase in baseline mucin secretion from KChIP3-depleted cells (*Figure 1B*), but there was no effect on agonist (ATP)-induced (stimulated) MUC5AC secretion (*Figure 1C*). Conversely, overexpression of KChIP3

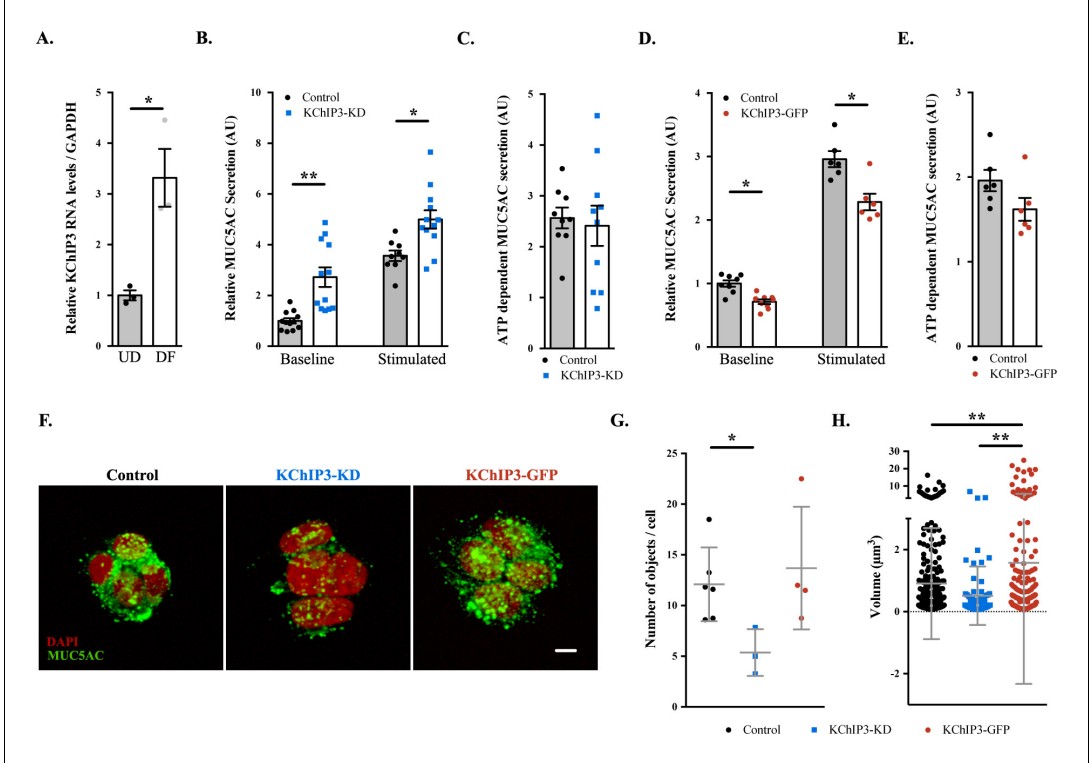

**Figure 1.** KChIP3 levels regulate baseline MUC5AC secretion. (**A**) KChIP3 RNA levels from undifferentiated (UD) and differentiated (DF) HT29-18N2 cells normalized by *GAPDH* values. (**B**) Control (black circles) and KChIP3 stable knockdown cells (KChIP3-KD) (blue squares) were differentiated and incubated for 30 min at 37°C in the absence or presence of 100 µM ATP. Secreted MUC5AC was collected and dot blotted with an anti-MUC5AC antibody. Data were normalized to actin levels. The y-axis represents normalized values relative to the values of untreated control cells. (**C**) ATP-dependent MUC5AC secretion was calculated from the data in (**B**) as the difference between normalized baseline secretion and stimulated secretion for each condition. (**D**) Secreted MUC5AC from differentiated control (black circles) and KChIP3 overexpressing cells (KChIP3-GFP) (red circles) in the absence or presence of 100 µM ATP. (**E**) ATP-dependent MUC5AC secretion calculated from the data in (**D**) for each condition. (**F**) Immunofluorescence Z-stack projections of control, KChIP3-KD and KChIP3-GFP differentiated HT29-18N2 cells with anti-MUC5AC antibody (green) and DAPI (red). Scale bar = 5 µm. (**G**) The number of MUC5AC granules for control (black circles), KChIP3-KD (blue squares) and KChIP3-GFP (red circles) cells was quantified from individual immunofluorescence stacks using 3D analysis FIJI software. The y-axis represents the number of 3-D objects detected by the software divided by the number of cells in each field. (**H**) Volume of control (black), KChIP3-KD (blue) and KChIP3-GFP (red) MUC5AC granules was calculated from individual immunofluorescence stacks using 3D analysis FIJI software. The y-axis represents the volume of the granules in µm$^3$. Abbreviations: UD: Undifferentiated HT29-18N2 cells, DF: Differentiated HT29-18N2 cells. *p<0.05, **p<0.01.

DOI: https://doi.org/10.7554/eLife.39729.002

The following figure supplements are available for figure 1:

**Figure supplement 1.** KChIP expression levels in HT29-18N2 stable cell lines.
DOI: https://doi.org/10.7554/eLife.39729.003

**Figure supplement 2.** KChIP3 levels regulate MUC2 secretion.
DOI: https://doi.org/10.7554/eLife.39729.004

(KChIP3-GFP cells) produced a 30% reduction in baseline MUC5AC secretion (*Figure 1D*), without affecting ATP-dependent MUC5AC secretion (*Figure 1E*).

MUC5AC secretion by colonic cancer cells is a good model system to study the mucin secretory pathway. Although MUC5AC is expressed at low levels in the gastrointestinal tract and upregulated in pathological conditions such as ulcerative colitis or parasitic infection (*Forgue-Lafitte et al., 2007*; *Hasnain et al., 2011*), under physiological conditions colonic goblet cells secrete MUC2. This raises the obvious question: is KChIP3 involved in baseline MUC2 secretion? We used the same procedure as described above to test MUC2 released into the medium in the absence of agonist, from differentiated KChIP3-KD, KChIP3-GFP and control HT29-18N2 cells. In accordance with our results with MUC5AC, KChIP3 levels significantly affected baseline MUC2 secretion. KChIP3-KD cells showed a 5.7-fold increase in baseline secretion compared to control cells, while KChIP3-GFP cells presented

a strong decrease (70.2% reduction compared to control cells) (*Figure 1—figure supplement 2A and B*). These findings indicate that the secretion of gel-forming mucins from colonic goblet cells might follow a similar mechanistic pathway. Even though the effects of MUC2 secretion are higher, for the sake of simplicity, availability and the cost of reagents, we have monitored the release of MUC5AC in the following experiments, unless otherwise mentioned.

Is the effect on MUC5AC secretion by KChIP3 depletion or overexpression perhaps due to changes in MUC5AC protein levels? Total cell lysate from KChIP3-KD, KChIP3-GFP and control cells was dot blotted with anti-MUC5AC antibody and we did not observe any obvious difference in the total MUC5AC intracellular levels, compared to control HT29-18N2 cells (*Figure 1—figure supplement 2C and D*). We then tested whether loss or overexpression of KChIP3 affected production of MUC5AC-containing secretory granules. Cells were imaged by confocal immunofluorescence microscopy with anti-MUC5AC antibody and analysed using FIJI software. Our results reveal that KChIP3-KD cells contained significantly fewer granules than control cells: 5.4 MUC5AC-containing particles/cell versus 13.7 particles/cell (p=*0.0239*), while the average size of the individual particles was not significantly reduced (0.90 $\mu m^3$ in control cells to 0.51 $\mu m^3$ in KChIP3-KD cells, p=*0.2430*). On the other hand, KChIP3-GFP overexpressing cells showed a dramatic accumulation of apical mucin granules, which was detected as an increase in the size of MUC5AC positive particles compared to control cells (1.60 $\mu m^3$ in KChIP3-GFP cells compared to 0.90 $\mu m^3$ in control cells, p=*0.0038*), however, we did not detect an appreciable change in the total number of granules (12.1 objects/cell in control *versus* 13.7 objects/cell in KChIP3-GFP cells, p=*0.6115*) (*Figure 1F*, quantification in *Figure 1G and H*).

Altogether, our data reveal that KChIP3 depletion increases the baseline secretion of MUC5AC and MUC2 as demonstrated by an increase in extracellular mucin with a concomitant reduction in the number of intracellular MUC5AC-containing granules. Conversely, KChIP3-GFP overexpressing cells secrete considerably less MUC5AC and present accumulation of apical MUC5AC granules.

## Spontaneous Ca$^{2+}$ oscillation-dependent mucin secretion by goblet cells

How does KChIP3 function in the secretion of MUC5AC and MUC2? Is it related to its ability to bind Ca$^{2+}$? To test this possibility, we evaluated the Ca$^{2+}$-dependency of baseline mucin secretion. We measured intracellular Ca$^{2+}$ levels with Fura-2AM dye in differentiated HT29-18N2 cells under baseline or stimulated secretion conditions (the absence or presence of physiological stimulus, ATP). Our data show that in the absence of stimulus, HT29-18N2 cells exhibit spontaneous intracellular Ca$^{2+}$ oscillations (average of 30% of cells in 10 min) (*Figure 2A*, left panel) of lower amplitude and shorter in time compared to an ATP-dependent (100 $\mu$M ATP) Ca$^{2+}$ response. In order to test the relevance of these oscillations to mucin secretion, we first identified the source of the Ca$^{2+}$, to enable us the possibility of manipulating these oscillations. 1) Removal of extracellular Ca$^{2+}$ (solution with 0.5 mM EGTA), increased the number of oscillating cells (96% of cells in 10 min) (*Figure 2A*, central panel) and 2) inhibition of ryanodine receptors (RYRs) (treatment with 10 $\mu$M dandrolene (*Zhao et al., 2001*)) reduced the number of Ca$^{2+}$ oscillations (13% of cells in 10 min) in either Ca$^{2+}$-containing extracellular solutions (*Figure 2A*, right panel and quantification of oscillations in *Figure 2B*) or in Ca$^{2+}$-free extracellular solutions (*Figure 2—figure supplement 1A and B*). These data implicate internal Ca$^{2+}$ stores (principally the ER) as the source of Ca$^{2+}$ oscillations in goblet cells. Importantly, our data show that RYRs are involved in the generation and maintenance of these oscillations.

Next, we tested whether intracellular Ca$^{2+}$ oscillations were involved in baseline mucin secretion. Briefly, differentiated goblet cells were incubated with vehicle, 0.5 mM EGTA or 10 $\mu$M dandrolene for 30 min at 37°C and MUC5AC secretion measured by the antibody-based dot blot procedure. Our results show that increased Ca$^{2+}$ oscillations (EGTA treatment) correlate with increased baseline mucin secretion (60% higher than control), while fewer Ca$^{2+}$ oscillations, in cells treated with 10 $\mu$M dandrolene, result in reduced baseline secretion (40% reduction) (*Figure 2C*). Altogether, these data suggest that intracellular Ca$^{2+}$ oscillations play an important role in baseline mucin secretion. This implies the involvement of a Ca$^{2+}$ sensor (*Adler et al., 2013*) that detects intracellular Ca$^{2+}$ oscillations to regulate baseline mucin secretion.

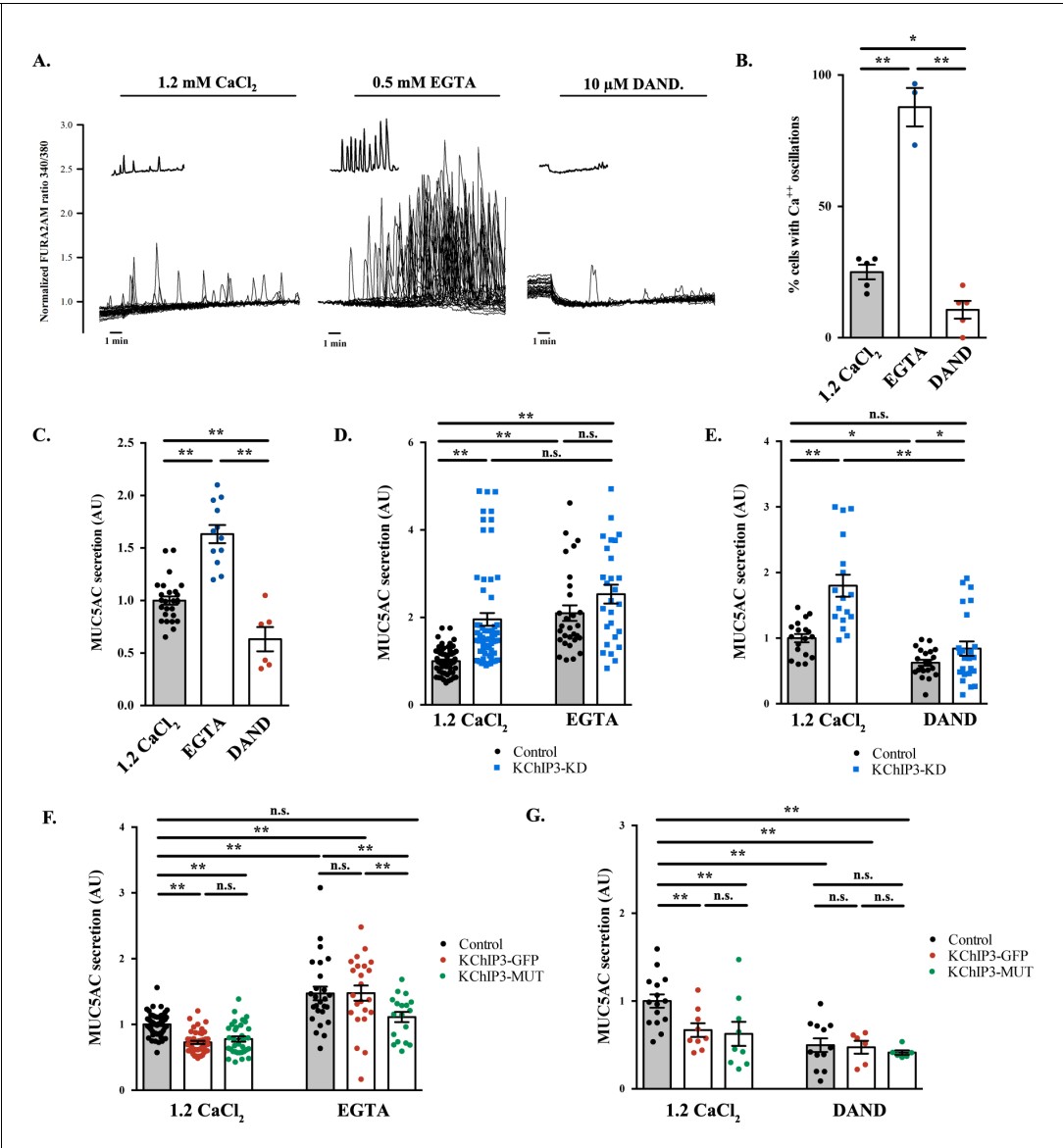

**Figure 2.** Ca²⁺ oscillations in goblet cells control KChIP3 function. (**A**) Time course of Ca²⁺ responses (normalized FURA-2AM ratio) obtained in differentiated HT29-18N2 cells in resting conditions exposed to different extracellular buffers: 1.2 mM CaCl₂ (left), 0.5 mM EGTA (center), or 10 µM dandrolene (right) (n = 30, inset shows a recording obtained from a single cell under each condition). (**B**) Percentage of cells oscillating in each condition during 10 min. Average values ± SEM are plotted as scatter plot with bar graph (N > 3) (black dots: 1.2 mM CaCl₂, blue dots: EGTA, red dots: dandrolene). (**C**) Secreted MUC5AC collected from differentiated HT29-18N2 cells that were incubated for 30 min at 37°C with different buffers: 1.2 mM CaCl₂ (black dots), 0.5 mM EGTA (blue dots) or 10 µM dandrolene (red dots). The y-axis represents relative values with respect to the values of control cells. Average values ± SEM are plotted as scatter plot with bar graph (N > 3). (**D**) Secreted MUC5AC from differentiated control (black circles) and KChIP3 stable knockdown cells (KChIP3-KD) (blue squares) collected after 30 min incubation at 37°C in the in the presence (1.2 mM CaCl₂) or absence (0.5 mM EGTA) of extracellular Ca²⁺. Data were normalized to intracellular actin levels. The y-axis represents normalized values relative to the values of untreated control cells. (**E**) Secreted MUC5AC from control (black circles) and KChIP3 stable knockdown cells (KChIP3-KD) (blue squares) that were incubated for 30 min at 37°C with vehicle or 10 µM dandrolene (DAND) in the presence of extracellular Ca²⁺. Data were normalized to intracellular actin levels. The y-axis represents normalized values relative to the values of untreated control cells. (**F**) Secreted MUC5AC from differentiated control (black circles), KChIP3-GFP (red circles) and KChIP3-MUT (green circles) cells that were incubated for 30 min at 37°C in the in the presence (1.2 mM CaCl₂) or absence (0.5 mM EGTA) of extracellular Ca²⁺. Data were normalized to intracellular actin levels. The y-axis represents normalized values relative to the values of untreated control cells. (**G**) Secreted MUC5AC from differentiated control (black circles), KChIP3-GFP (red circles) and KChIP3-MUT (green circles) cells after 30 min incubation at 37°C with vehicle or 10 µM dandrolene (DAND) in the presence of extracellular Ca²⁺. Data were normalized to intracellular actin levels. The y-axis represents normalized values relative to the values of untreated control cells. Abbreviations: EGTA: Buffer with 0.5 mM EGTA, DAND: 10 µM Dandrolene treatment. *p<0.05, **p<0.01, n.s.: not statistically significant.

DOI: https://doi.org/10.7554/eLife.39729.005

*Figure 2 continued on next page*

*Figure 2 continued*

The following figure supplement is available for figure 2:

**Figure supplement 1.** KChIP3 does not control Ca$^{2+}$ levels.

DOI: https://doi.org/10.7554/eLife.39729.006

## KChIP3 function is regulated by intracellular Ca$^{2+}$ oscillations

Our data on KChIP3 and RYR-dependent Ca$^{2+}$ oscillations in mucin secretion and a recent description of a functional interaction between KChIP3 and neuronal RYR (*Grillo et al., 2018*), led us to test whether KChIP3 was the link between Ca$^{2+}$ oscillations and mucin secretion.

We considered two possibilities: Ca$^{2+}$ oscillations control KChIP3 activity to regulate MUC5AC secretion or KChIP3 affects MUC5AC baseline secretion by controlling Ca$^{2+}$ oscillations. To distinguish between these two possibilities, we first tested whether Ca$^{2+}$ oscillations regulate KChIP3's effect on baseline mucin secretion.

Our results show that increasing Ca$^{2+}$ oscillations by removing extracellular Ca$^{2+}$ (EGTA solution) increased MUC5AC secretion in control but not in KChIP3-KD cells (*Figure 2D*). Furthermore, EGTA treatment abolished the differences in mucin secretion between control and KChIP3-KD cells (2.1 *vs.* 2.5 fold increase, respectively), suggesting that removal of KChIP3 brings cells close to their maximal baseline mucin secretion. Additionally, decreasing the number of Ca$^{2+}$ oscillations (dandrolene treatment) equally reduced baseline mucin secretion in both control and KChIP3-KD cells (*Figure 2E*), suggesting that intracellular Ca$^{2+}$ oscillations are key to baseline mucin secretion and that in the absence of these Ca$^{2+}$ signals, KChIP3 disengages its function as modulator of baseline mucin secretion.

Second, to test whether the link between KChIP3 and Ca$^{2+}$ oscillations to regulate baseline mucin secretion relates to the Ca$^{2+}$ binding capability of KChIP3 we generated a stable HT29-18N2 cell line overexpressing an EF-hand mutant KChIP3 (KChIP3-MUT), which is unable to bind Ca$^{2+}$ (*Carrión et al., 1999*) (expression levels were tested by western blot, as shown in *Figure 1—figure supplement 1B*). Under normal basal Ca$^{2+}$ conditions (1.2 mM CaCl$_2$), differentiated KChIP3-MUT cells showed a similar reduction in baseline MUC5AC (*Figure 2F*) and MUC2 secretion (*Figure 1—figure supplement 2B*) as KChIP3-GFP cells (37% and 47% decrease compared to control, respectively) compared to control cells. However, increasing intracellular Ca$^{2+}$ oscillations (0.5 mM EGTA treatment) induced different behaviour of WT and mutant KChIP3-expressing cells. While KChIP3-GFP cells showed increased secretion of MUC5AC identical to control cells (47.1% increase in control cells *vs.* 47.6% in KChIP3-GFP cells), KChIP3-MUT cells only showed an 11.1% increased secretion (*Figure 2F*). On the other hand, reducing intracellular Ca$^{2+}$ oscillations with dandrolene treatment markedly reduced MUC5AC secretion regardless of the form of KChIP3 overexpressed (*Figure 2G*).

Together, these observations suggest that intracellular Ca$^{2+}$ oscillations under control- unstimulated- conditions regulate baseline mucin secretion and that KChIP3 acts as a brake for mucin secretion. Knocking down KChIP3 increases while overexpression of KChIP3 decreases baseline mucin secretion. In addition, the role of KChIP3 in mucin secretion depends on basal intracellular Ca$^{2+}$ signals and the ability of KChIP3 to sense such Ca$^{2+}$ signals. Under conditions of low expression of KChIP3 (therefore a reduced brake capability and, consequently higher baseline mucin secretion), there is no further effect on secretion even with an increase in Ca$^{2+}$ oscillations. On the contrary, overexpression of KChIP3 (increased brake capability) inhibits secretion and this effect is reverted by increasing Ca$^{2+}$ oscillations, mainly in cells overexpressing KChIP3-GFP, but not KChIP3-MUT. We therefore conclude that both Ca$^{2+}$ oscillations and KChIP3 function in the same pathway of baseline mucin secretion and Ca$^{2+}$ oscillations likely control KChIP3 function. However, to completely discard the second possibility raised above, we tested whether KChIP3 had a role in the generation or maintenance of Ca$^{2+}$ oscillations.

We measured intracellular Ca$^{2+}$ levels by Fura2-AM in HT29-18N2 cells depleted of KChIP3 (KChIP3-KD) or cells overexpressing KChIP3 (KChIP3-GFP) under baseline (15 min in 1.2 mM CaCl$_2$) or stimulated conditions (10 min, 1.2 mM CaCl$_2$, 100 μM ATP). Our results show no difference in basal Ca$^{2+}$ levels (*Figure 2—figure supplement 1C and D*), ATP-dependent Ca$^{2+}$ entry (*Figure 2—figure supplement 1E and F*), or in the number of cells exhibiting Ca$^{2+}$ oscillations in KChIP3-KD or

KChIP3-GFP cells compared to control cells (33% in KChIP3-KD, 37% in KChIP3-GFP, 30% in control cells) (*Figure 2—figure supplement 1G*). Thus, we conclude that KChIP3 has no role in the generation of spontaneous intracellular $Ca^{2+}$ oscillations or ATP-mediated $Ca^{2+}$ entry in HT29-18N2 goblet cells.

Altogether, these results suggest that $Ca^{2+}$ oscillations generated in goblet cells are sensed by KChIP3 to control baseline mucin secretion. We further propose that KChIP3 in its $Ca^{2+}$-free form acts as a repressor for baseline mucin secretion.

## KChIP3 localizes to a pool of mucin secretory granules

Based on the cell type and specific post-translational modifications including palmitoylation, sumoylation and GRK-dependent phosphorylation, KChIP3 is reported to be cytoplasmic, nuclear and/or located at the plasma membrane (*Palczewska et al., 2011*; *Ruiz-Gomez et al., 2007*; *Takimoto et al., 2002*; *Zaidi et al., 2004*). However, the intracellular localization of KChIP3 in non-excitable goblet cells has not been described before. We therefore tested the location of KChIP3 in differentiated HT29-18N2 cells (*Figure 3*). KChIP3-GFP cells and KChIP3-MUT cells were differentiated and seeded on coverslips. KChIP3-GFP and KChIP3-MUT cells were first permeabilized and

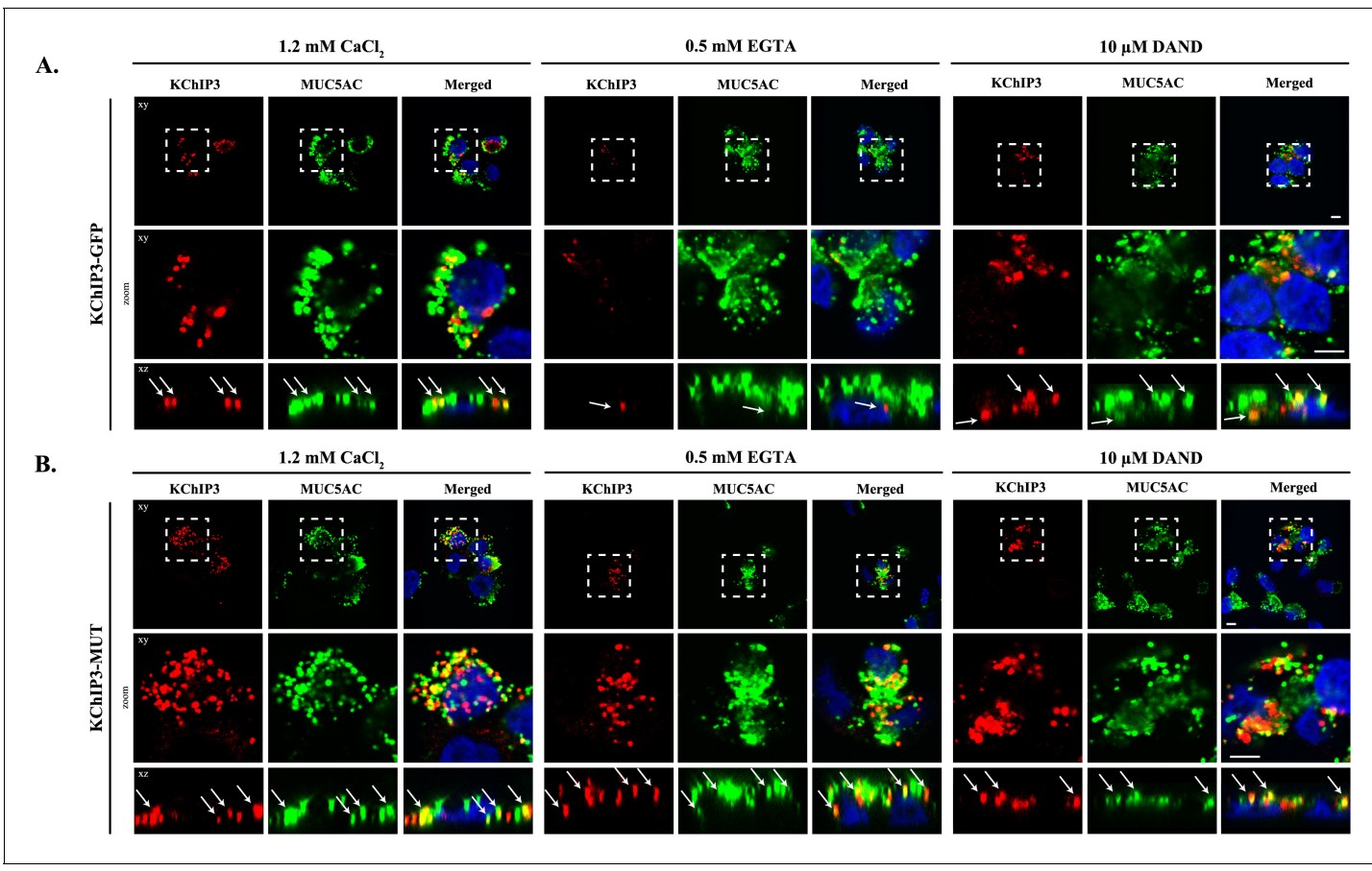

**Figure 3.** KChIP3 localized to a pool of MUC5AC granules. Differentiated KChIP3-GFP (**A**) and KChIP3-MUT (**B**) cells were processed for cytosolic washout after 30 min at 37°C of treatment with 1.2 mM $CaCl_2$, 0.5 mM EGTA or 10 µM dandrolene. After fixation and permeabilization, samples were analyzed by immunofluorescence microscopy with an anti-GFP (KChIP3, red), anti-MUC5AC antibody (MUC5AC, green) and DAPI (blue). Images represent a single plane (xy), a zoom of the area within the white square and an orthogonal view of each channel (xz). Scale bar = 5 µm. White arrows point to the colocalization between KChIP3 and MUC5AC. Abbreviations: EGTA: Buffer with 0.5 mM EGTA, DAND: 10 µM dandrolene treatment.
DOI: https://doi.org/10.7554/eLife.39729.007

The following figure supplement is available for figure 3:

**Figure supplement 1.** KChIP3 apical localization in HT29-18N2 goblet cells.
DOI: https://doi.org/10.7554/eLife.39729.008

washed extensively to remove the soluble cytoplasmic pool of KChIP3, thereby facilitating visualization of KChIP3 associated to the cytoplasmic face of intracellular compartments. Immunofluorescence microscopy with anti-KChIP3 antibody and anti-GFP antibody revealed presence of KChIP3-GFP and KChIP3-MUT on apical elements (*Figure 3—figure supplement 1A*, and co-localization between GFP and KChIP3 is shown in *Figure 3—figure supplement 1B*). KChIP3-containing *punctae* show significant colocalization with an apical pool of MUC5AC-containing granules. Our data reveal that 10 – 15% of total MUC5AC granules, detected by immunofluorescence (IF), contain KChIP3-GFP (*Figure 3—figure supplement 1C*, upper panel). Quantification of the images (using FIJI software, and described in materials and methods) revealed that in KChIP3-GFP cells under basal conditions, 40.6% of KChIP3 signal colocalized with MUC5AC (Average Manders' coefficient = 0.4061) (*Figure 3—figure supplement 1D*). In accordance with the functional data presented in the previous section, KChIP3-MUT showed very similar localization to KChIP3-GFP and even higher colocalization with MUC5AC granules (59% of KChIP3-MUT signal colocalizing with MUC5AC under basal conditions) (*Figure 3—figure supplement 1C*, lower panel; quantification in *Figure 3—figure supplement 1F*). These data suggests that KChIP3 associates with a pool of mucin granules in a $Ca^{2+}$-free form.

We then tested whether localization of KChIP3-GFP and KChIP3-MUT to the apical pool of MUC5AC-containing granules depends on intracellular $Ca^{2+}$ oscillations. We analysed KChIP3 localization after 30 min at 37°C under normal conditions (vehicle, 1.2 mM $CaCl_2$), increased $Ca^{2+}$ oscillations (solution with 0.5 mM EGTA), or decreased oscillations (10 µM dandrolene treatment). Our data show that the percentage of KChIP3 colocalizing with MUC5AC granules remains stable over time (KChIP3-GFP: 41% at t = 0 *vs.* 46% at t = 30', *Figure 3A* left panel, KChIP3-MUT: 59% at t = 0 *vs.* 51% at t = 30', *Figure 3B* left panel) under normal conditions. Interestingly, increasing the number of the oscillations by EGTA treatment for 30 min at 37°C, reduced the number of KChIP3-GFP positive granules (*Figure 3A*, central panel) (quantified in *Figure 3—figure supplement 1E* as the volume of KChIP3 signal, 12 µm$^3$ *vs.* 4.4 µm$^3$, respectively) and colocalization with MUC5AC (34% control *vs.* 19% EGTA treated cells), but it did not have any effect on KChIP3-MUT localization to granules (*Figure 3B* central panel) (52% in control *vs.* 54% in EGTA treated cells colocalization with MUC5AC, *Figure 3—figure supplement 1F*) or the number of KChIP3 granules (12.4 *vs.* 12.3 µm$^3$, *Figure 3—figure supplement 1G*). Finally, reducing the number of $Ca^{2+}$ peaks did not significantly affect the number of KChIP3 accumulations or colocalization to MUC5AC granules in KChIP3-GFP (*Figure 3A* right panel, quantifications in *Figure 3—figure supplement 1D and E*) or KChIP3-MUT cells (*Figure 3B*, quantification in *Figure 3—figure supplement 1F and G*).

In sum, our data show that intracellular $Ca^{2+}$ oscillations regulate KChIP3 localization at MUC5AC granules: Increase in $Ca^{2+}$ oscillations reduces KChIP3 localization to MUC5AC granules, whereas impairing $Ca^{2+}$ binding to KChIP3 results in more KChIP3-containing MUC5AC granules.

## *Kcnip3*$^{-/-}$ mice show increased colonic mucus layer under basal conditions

KChIP3, encoded by the gene *Kcnip3*, is expressed in the colon of wild type (WT) mice and its deletion (*Kcnip3*$^{-/-}$ mice) does not affect the levels of the other members of the family (*Figure 4—figure supplement 1A*). *Kcnip3*$^{-/-}$ mice are reported to exhibit decreased chronic neuropathic or inflammatory acute pain behaviours, without any major defects in locomotion, learning or memory (*Cheng et al., 2002*). However, other effects of KChIP3 loss on the overall mouse physiology are not known. Based on our data showing that KChIP3 controls baseline mucin secretion from colonic goblet cells, we tested the effects of KChIP3 deletion on the levels of mucins in the colon-specific mucus layer under -non stimulatory- baseline conditions.

Colons were extracted from 12-week-old WT and *Kcnip3*$^{-/-}$ mice (N = 6 per genotype), fixed by the standard methanol-carnoy method (as described in (*Johansson and Hansson, 2012*)) and stained with Periodic acid–Schiff (PAS) or Periodic acid–Schiff - Alcian blue (PAS-AB) to detect acidic and neutral mucins in the medial and distal colon. The PAS staining procedure is also used to detect glycogen and other glycoproteins, however, combination of both PAS and Alcian-Blue is the most sensitive method to detect mucins because these reagents detect all mucins regardless of their charge (*Mantle and Allen, 1978*). Impressively, distal colon of *Kcnip3*$^{-/-}$ mice revealed significantly increased levels of mucus layer compared to WT mice for both PAS (43.0 µm control *vs.* 77.0 µm *Kcnip3*$^{-/-}$) (*Figure 4A and B*) and PAS-AB (40.5 µm in WT mice *vs.* 73.6 µm in *Kcnip3*$^{-/-}$ mice)

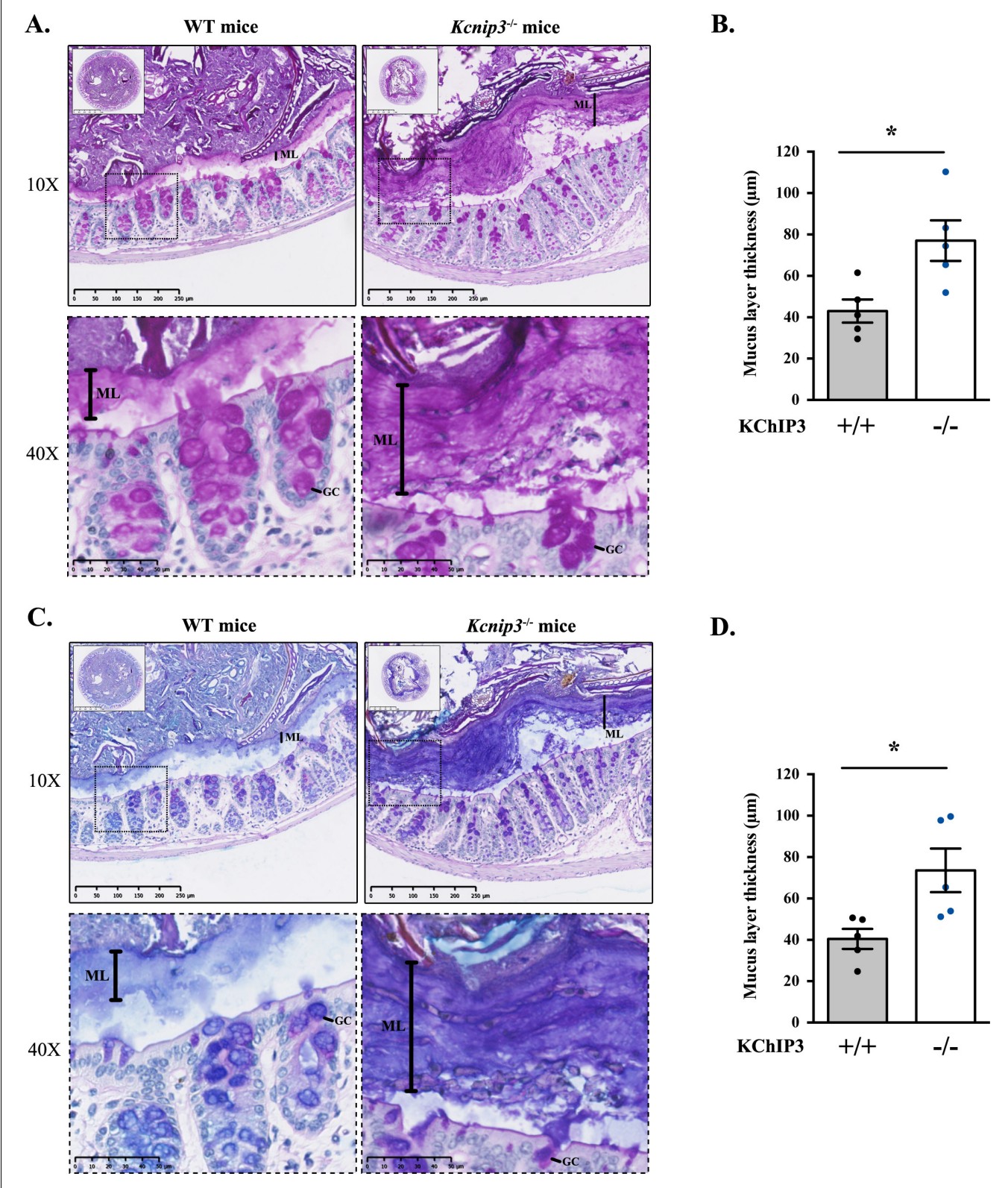

**Figure 4.** *Kcnip3⁻/⁻* mice show increased mucus layer thickness. (A, C) Representative distal colons of WT (left panel) and *Kcnip3⁻/⁻* (right panel) mice stained with PAS (A) or PAS-AB (C) at different magnification (2.5X, 10X and 40X). (B, D) Quantification of the mucus layer thickness in the distal colon stained with PAS (B) or PAS-AB (D) of WT (black dots) and *Kcnip3⁻/⁻* (blue dots) mice. Average values ± SEM are plotted as scatter plot with bar graph.

*Figure 4 continued on next page*

*Figure 4 continued*

The y-axis represents the thickness of the mucus layer in μm. Abbreviations: +/+: WT mice, -/-: *Kcnip3*[-/-] mice, ML: Mucus layer, GC: Goblet cell.
*p<0.05.
DOI: https://doi.org/10.7554/eLife.39729.009

The following figure supplement is available for figure 4:

**Figure supplement 1.** *Kcnip3*[-/-] mice show increased mucus layer at the medial colon.
DOI: https://doi.org/10.7554/eLife.39729.010

(*Figure 4C and D*). In addition, this increase in the mucus layer was also detected in the medial colon of four out of five *Kcnip3*[-/-] mice (average thickness, PAS: 31.6 μm WT mice *vs.* 50.3 μm *Kcnip3*[-/-] mice, NS; PAS-AB: 33.6 μm WT mice *vs.* 54.0 μm *Kcnip3*[-/-] mice, NS) (*Figure 4—figure supplement 1B–D*).

## Discussion

Homeostatic control of cellular protein secretion is essential, as evinced by the effects of dysregulated secretion of neurotransmitters, cytokines, insulin, hormones and mucins on organismal physiology. Specialized goblet cells release heavily glycosylated mucins from stored granules via exogenously applied agonist-dependent and -independent pathways. The former pathway is a form of acute response to protect epithelial lining when exposed to pathogens or allergens. The agonist-independent pathway however, is for chronic, continuous release of mucins at a low rate. Recent studies suggest that baseline secretion can release mucins in quantities that exceed several-fold, the amounts released by the agonist-mediated process over a 24 hr period (*Zhu et al., 2015*). While both of these pathways are poorly studied, our understanding of how cells control baseline mucin secretion is relatively even more primitive. In a nutshell, all that is known at present about baseline mucin secretion in human bronchial epithelial cells and mouse trachea is the involvement of SNAP23 and Munc13-2, which also regulate stimulated secretion (*Ren et al., 2015*; *Zhu et al., 2008*). It is also known that baseline mucin secretion is independent of Syt2 and sensitive to mechanical stress (*Zhu et al., 2015*). But how is baseline mucin secretion controlled to prevent pathological quantities of secreted mucins?

Our data reveal a linear pathway whereby spontaneous cellular $Ca^{2+}$ oscillations trigger KChIP3 detachment from mucins granules and promote mucin secretion in an agonist-independent manner. We propose KChIP3 as the high-affinity calcium sensor to control baseline mucin secretion. KChIP3 according to our data is a negative regulator of baseline mucin secretion. Syt2, on the other hand, is the low-affinity calcium sensor that acts positively to regulate stimulated mucin secretion (*Tuvim et al., 2009*). Why is baseline mucin secretion controlled by a negative regulator? One possibility is that mature granules for baseline secretion are already docked at plasma membrane and ready to release their contents. The only means, therefore, to prevent their fusion is by employing a brake –a negative regulator– such as KChIP3. Another possibility that we cannot exclude is that KChIP3 competes with a positive regulator, like synaptotagmins, to control baseline mucin secretion.

In addition, we propose that even though there are notable differences in the proposed mechanisms of mucin secretion between colonic and airway cells, the function of KChIP3 is conserved in mucin secreting goblet cells regardless of the tissue. For example, both stimulated and baseline secretion in the airways are affected by shear stress (airflow and mucus movements promoted by ciliated cells), but are independent of extracellular $Ca^{2+}$ (*Davis and Dickey, 2008*; *Tuvim et al., 2009*). In the intestine, there are two layers of mucus, an outer loose layer and an inner adherent layer of MUC2 (which is free of pathogens) (*Johansson et al., 2011*), and extracellular $Ca^{2+}$ is required for stimulated secretion (*Mitrovic et al., 2013*). Nevertheless, mucin secretion in both airways and the colon requires intracellular $Ca^{2+}$. Altogether, this suggests the existence of a low affinity $Ca^{2+}$ sensor for stimulated secretion and a high affinity $Ca^{2+}$ sensor for baseline secretion regardless of the tissue type. KChIP3, as the high affinity $Ca^{2+}$ sensor, therefore likely plays an important role in the physiology and pathophysiology of colon, the airways and consequently in mucin related pathologies such as asthma, cystic fibrosis and COPD.

## Ca²⁺ oscillations control baseline mucin secretion in colonic cells

The first key, novel data presented here are the exhibition of spontaneous $Ca^{2+}$ oscillations in goblet cells. These oscillations arise from $Ca^{2+}$ release from internal stores (mainly ER), in a ryanodine receptor (RYR)-dependent manner. By modulating the frequency of $Ca^{2+}$ oscillations, we could perturb the release propensity of mucin granules in goblet cells. It is known that RYRs promote $Ca^{2+}$ release from the ER, which is necessary for several $Ca^{2+}$-dependent intracellular functions (*Van Petegem, 2012*). Our findings reveal an important new function of these intracellular $Ca^{2+}$ oscillations as a means to control quantities of mucin secreted by goblet cells. We propose that goblet cells exhibit regulated secretory activity in two different regimes of intracellular $Ca^{2+}$: 1) a steady, moderated release (baseline) using intracellular stores to mildly elevate $Ca^{2+}$ levels (oscillations) and, 2) a burst of secretion (stimulated) in response to potent physiological stimuli (ATP, for example) or pathogenic insults (for instance, allergenic) with a sizable influx of extracellular $Ca^{2+}$. This also helps resolve the controversy on the source of $Ca^{2+}$ for mucin release: extracellular $Ca^{2+}$ is used to control stimulated release, whereas intracellular $Ca^{2+}$ is employed to manage baseline mucin secretion.

## KChIP3 links Ca²⁺ oscillations to mucin secretion

In neurons, KChIP3 alters ER calcium content and RYR-mediated $Ca^{2+}$-induced $Ca^{2+}$ release (CICR) by direct interaction with RYR receptors (*Grillo et al., 2018*; *Lilliehook et al., 2002*). As shown here, in HT29-18N2 goblet cells, modulating KChIP3 levels did not perturb intracellular $Ca^{2+}$ homeostasis, but the localization of KChIP3 to granules is controlled by $Ca^{2+}$ oscillations. It is reported that ER is in close proximity to mucin granules in goblet cells (*Tuvim et al., 2009*), which presents the possibility that by directly interacting with RYR, the activity and location of KChIP3 is affected by an ER-based $Ca^{2+}$ oscillations. KChIP3 in this manner couples $Ca^{2+}$ release from ER to baseline mucin secretion. Thus, the second discovery presented here is that KChIP3 localization to granules and its function is linked to $Ca^{2+}$ oscillations. Increasing $Ca^{2+}$ oscillation frequency reduces the fraction of mucin granules bearing KChIP3, which correlates with an increase in baseline mucin secretion. Furthermore, overexpressing KChIP3 or a mutant KChIP3 that cannot bind $Ca^{2+}$ reduces baseline mucin secretion. Loss of KChIP3, on the other hand, creates mucin hypersecretion. Based on our findings, we suggest that $Ca^{2+}$-bound KChIP3 is soluble and cytoplasmic: under these conditions the cells release more mucins from the secretory granules. This hypersecretory phenotype is mimicked by cells having more $Ca^{2+}$ oscillations, with a concomitant decrease in the number of KChIP3 containing granules. Cells overexpressing KChIP3-GFP on the other hand release less mucins and accumulate apical mucin granules. It could be argued that modulation of calcium oscillations changes mucin granule' composition and this somehow affects KChIP3 localization. This is unlikely because the location of KChIP3-MUT, which cannot bind $Ca^{2+}$, is unaffected by perturbing $Ca^{2+}$ oscillations. Whereas, the wild type KChIP3-GFP location changes as per the status of the $Ca^{2+}$ oscillations. Also, the fast kinetics of the recruitment and release of KChIP3 under our experimental conditions strongly suggest that the changes in KChIP3 location presented here are unlikely due to in changes in granule composition.

   How does the attachment of KChIP3 to the granules inhibit their release propensity? We suggest a parsimonious model, that granule-bound KChIP3 inhibits events leading to SNARE-mediated fusion. Whether this is by sequestering $Ca^{2+}$, competing with the proteins of the SNARE complex or with regulators of the SNARE assembly such as Munc13-2, is an important new challenge. It is known that proteins like synaptotagmins act as low affinity $Ca^{2+}$ sensor in the agonist-dependent release at the nerve terminal and in non-excitable goblet cells. KChIP3, we propose, is the high affinity $Ca^{2+}$ sensor that controls release propensity of mucin granules in non-excitable goblet cells in the absence of stimuli. It is possible that KChIP3 also has a role at the nerve terminal and in the extracellular $Ca^{2+}$ independent release of cargoes that are stored in secretory granules.

   Altogether, we suggest that $Ca^{2+}$ oscillations generated in the proximity of mucin granules from the ER, and not small increases in mean cytoplasmic calcium concentration, reach a certain threshold (provided by KChIP3's $Ca^{2+}$ binding properties) to trigger baseline mucin secretion. We hypothesize that binding of $Ca^{2+}$ to KChIP3 changes its conformation thereby affecting its interaction to mucin granules. This, we suggest, is analogous to KChIP3's association to DRE sites where it acts as a transcriptional repressor. In this scheme, KChIP3 bound to $Ca^{2+}$ dissociates from DRE sites (*Carrión et al., 1999*). However, further work is needed to assess whether this proposed $Ca^{2+}$

dependent conformational change that triggers dissociation from the granules is due to changes in the binding affinity of KChIP3 to granules or a change in its binding partners at the granules.

## Loss of KChIP3 causes mucin hypersecretion in vivo

It is satisfying to note that the effects of loss of KChIP3 on mucin hypersecretion by colonic goblet cells in culture are replicated in the colon of *Kcnip3*[-/-] mouse. Both in the medial and distal colons of 12 week old *Kcnip3*[-/-] mice, there is a clear increase in the mucus layer thickness, from an average of 40 μm in wild type to 80 μm in *Kcnip3*[-/-] mice. However, the increased extracellular pool of mucins in *Kcnip3*[-/-] mouse in the images shown might be perceived not to correlate to a decrease in intracellular mucin content within the goblet cells. The likely reason for this apparent disconnect is our inability to distinguish between mucins secreted into the colonic crypt and the intracellular mucins in the colonic epithelium. This can only be addressed reliably in cells isolated from the colon of the wild type and *Kcnip3*[-/-] mouse followed by separating extracellular pool of mucins from the cells, and subsequently quantitating the intracellular and extracellular pool by dot blotting as we have shown here for the colonic cancer cells. This is technically extremely challenging, but regardless of this potential shortcoming, the data clearly point to a significant increase in the extracellular pool of colonic mucins in *Kcnip3*[-/-] mouse compared to their wild type counterparts. Additionally, the colonic mucus layer in *Kcnip3*[-/-] mice seems denser (*Figure 4*), suggesting that the viscoelastic properties of the gel-forming mucins are also altered. Accumulated mucin in *Kcnip3*[-/-] mice colon creates a more compact mucous, which is likely to be detrimental for overall gut physiology.

## KChIP3: a mark of maturity

Based on the data presented here, we propose the following model for the regulation of mucin granule secretion (*Figure 5*).

1. Granule biogenesis and maturation: Heavily glycosylated mucins are sorted and packaged into micron-sized granules at the TGN. These granules undergo a number of maturation steps including condensation of mucins by a $Ca^{2+}$-dependent process (*Perez-Vilar, 2007*). How mucins are sorted and packed into granules, or how granules are generated from the Golgi, remains unknown.

2. There are no specific markers of the granules to identify and distinguish different stages of their maturity. We propose that membrane-anchored KChIP3 defines a mature subset of granules that are stalled in events leading to fusion. A RYR-mediated $Ca^{2+}$ spike increases local cytoplasmic $Ca^{2+}$ concentration that results in binding of $Ca^{2+}$ to KChIP3 followed by its detachment from the granules. The dissociation of KChIP3-$Ca^{2+}$ from granules disengages the brake holding granules from fusion, thereby resulting in mucin release. Spontaneous $Ca^{2+}$ oscillations are unlikely to be of the amplitude to involve low-affinity $Ca^{2+}$ binding sensor like the Syt2. KChIP3 might then be the key calcium sensor that functions instead of proteins like synaptotagmins to control baseline mucin secretion.

3. Physiological or pathological stimuli that result in a massive increase in intracellular $Ca^{2+}$ involve Syt2, which is a low-affinity $Ca^{2+}$ sensor to control release propensity of the granules (*Tuvim et al., 2009*). In this regime, we propose that KChIP3 is in $Ca^{2+}$-bound form and detached from mature granules.

In conclusion, specialized secretory cells based on the source of $Ca^{2+}$ engage different $Ca^{2+}$ binding proteins to release cargoes from the secretory granules. The agonist and extracellular $Ca^{2+}$-dependent fast release employs low-affinity $Ca^{2+}$ sensors like Syt2, whereas the agonist-independent and intracellular $Ca^{2+}$ oscillations exploit the function of a high-affinity $Ca^{2+}$ sensor like KChIP3. In the last three decades, we have gathered a detailed understanding of synaptotagmin's function as a clamp for fast $Ca^{2+}$ evoked release. The challenge is to unravel the mechanism by which KChIP3 permits the quantities of mucins released, and likely many other cargoes, to balance the physiological needs of an organism.

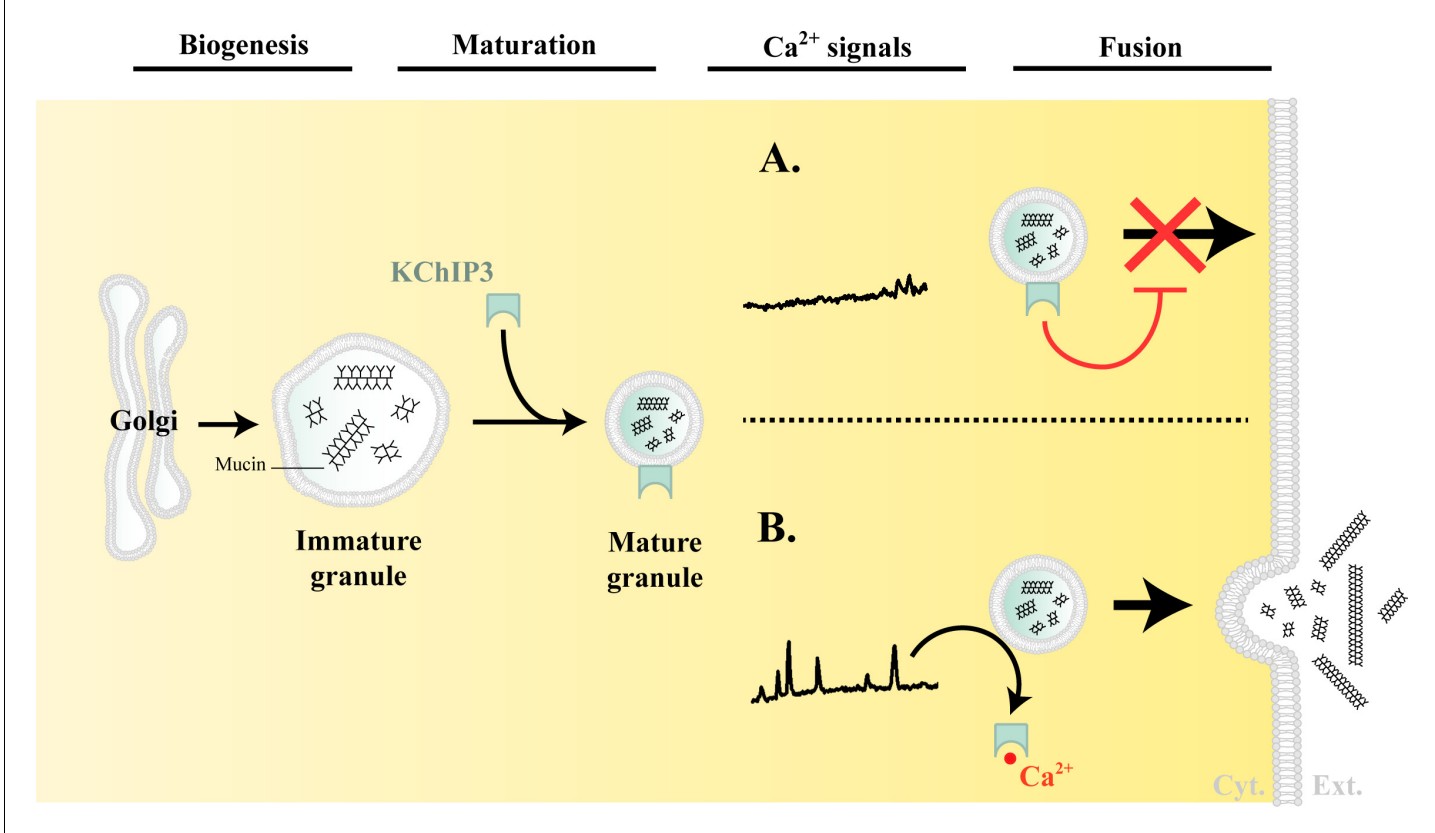

**Figure 5.** Model of KChIP3 function. Post biogenesis at the Golgi, mucin granules undergo maturation and are placed on path to fuse with the plasma membrane and release their contents. However, the mature granules recruit KChIP3 (unbound to $Ca^{2+}$) and this event stalls their fusion to the plasma membrane. KChIP3 recruitment thus acts as a brake to control the number of granules that can fuse with the plasma membrane. Based on our findings we describe two situations: (A) In absence of intracellular $Ca^{2+}$ oscillations, KChIP3 is bound to the granules preventing their secretion, and (B) Upon a rise in intracellular $Ca^{2+}$ by RYR-induced $Ca^{2+}$ oscillations, the KChIP3 on the granules binds $Ca^{2+}$ and subsequently detaches thereby triggering their SNARE dependent fusion to the plasma membrane. The cells have therefore evolved a mechanism by imposing the recruitment of empty KChIP3 to mucin granules to stall their progression leading to fusion with plasma membrane. Abbreviations: Cyt.: cytosol, Ext.: extracellular medium.
DOI: https://doi.org/10.7554/eLife.39729.011

## Materials and methods

### Reagents and antibodies
All chemicals were obtained from Sigma-Aldrich (St. Louis, MO) except anti-MUC2 antibody clone 996/1 (RRID:AB_297837) (Abcam, Cambridge, UK), anti-MUC5AC antibody clone 45M1 (RRID: AB_934745) (Neomarkers, Waltham, MA) and anti-KChIP3 antibody (RRID:AB_10608850) (Santa Cruz Biotechnology, Texas, USA). Secondary antibodies for immunofluorescence microscopy and dot blotting were from Life Technologies (ThermoFisher Scientific, Waltham, MA, USA).

### Cell lines
HT29-18N2 cells (obtained from ATCC) (RRID:CVCL_5942) were tested for mycoplasma contamination with the Lookout mycoplasma PCR detection kit (Sigma-Aldrich, St. Louis, MO). Mycoplasma negative HT29-18N2 were used for the experiments presented here.

### Generation of stable cell lines (shRNA and overexpression)
HEK293T cells (ATCC, negative for mycoplasma) were co-transfected with the plasmid, VSV-G, pPRE (packaging) and REV by $Ca^{2+}$ phosphate to produce lentiviruses. 48 hr post transfection, the secreted lentivirus was collected, filtered and directly added to HT29-18N2 (HT29-18N2 full

description as above) cells. Stably infected HT29-18N2 cells with the different constructs were sorted for GFP signal by FACS.

## qPCR

Differentiated HT29-18N2 control, KChIP3-KD, KChIP3-GFP, and KChIP3-MUT cells were lysed and total RNA extracted with the RNeasy extraction kit (Qiagen, Netherlands). cDNA was synthesized with Superscript III (Invitrogen). Primers for each gene (sequence shown below, *Table 1*) were designed using Primer-BLAST (NCBI) (*Ye et al., 2012*), limiting the target size to 150 bp and the annealing temperature to 60°C. To determine expression levels of KChIP1, KChIP2, KChIP3 and KChIP4, quantitative real-time PCR was performed with Light Cycler 480 SYBR Green I Master (Roche, Switzerland) according to manufacturer's instructions.

## Differentiation of HT29-18N2 cells

HT29-18N2 cells were differentiated to goblet cells as described previously (*Mitrovic et al., 2013*). Briefly, cells were seeded in complete growth medium (DMEM complemented with 10% FCS and 1% P/S), and the following day (Day zero: D-0), the cells were placed in PFHMII protein free medium (GIBCO, ThermoFisher Scientific, Waltham, MA, USA). After 3 days (D-3), medium was replaced with fresh PFHMII medium and cells grown for 3 additional days. At day 6 (D-6) cells were trypsinized and seeded for the respective experiments in complete growth medium followed by incubation in PFHMII medium at day 7 (D-7). All experimental procedures were performed at day 9 (D-9).

## Mucin secretion assay for HT29-18N2 cells

HT29-18N2 cells were differentiated for 6 days and then split into 6-well plates. After one day (D-7), medium was exchanged with fresh PFHMII medium and cells grown for 2 more days. On D-9, cells were washed with isotonic solution containing: 140 mM NaCl, 2.5 mM KCl, 1.2 mM $CaCl_2$, 0.5 mM $MgCl_2$, 5 mM glucose, and 10 mM HEPES (305 mosmol/l, pH 7.4 adjusted with Tris); and then treated with vehicle (baseline secretion) or 100 µM ATP (stimulated secretion) for 30 min at 37°C. $Ca^{2+}$-free solutions were obtained by replacing $CaCl_2$ with equal amounts of $MgCl_2$ and 0.5 mM EGTA. In order to inhibit RYR, cells were treated for 10 min with 10 µM dandrolene prior to start the secretion assay. During the secretion assay, cells were treated either with the drug or the respective vehicle. After 30 min at 37°C, extracellular medium was collected and centrifuged for 5 min at 800xg at 4°C. Cells were washed 2X in PBS and lysed in 2% Triton X-100/PBS for 1 hr at 4°C and centrifuged at 14000xg for 15 min.

## Dot blot analysis

Extracellular medium and cell lysates were spotted on nitrocellulose membranes (0.45 µm) using Bio-Dot Microfiltration Apparatus (Bio-Rad, California, USA) (manufacturer's protocol) and membranes were incubated in blocking solution (5% BSA/0.1% Tween/PBS) for 1 hr at room temperature. The blocking solution was removed and the membranes were incubated with an anti-MUC5AC antibody diluted 1:2000 or the anti-MUC2 antibody diluted 1:4000 in blocking solution, overnight at 4°C. Membranes were then washed in 0.1% Tween/PBS and incubated with a donkey anti-mouse or anti-rabbit HRP coupled antibody (Life Technologies) for 1 hr at room temperature. For the detection of ß-actin and KChIP3, cell lysates were separated on SDS-PAGE, transferred to nitrocellulose

**Table 1.** Primer sequences used for detecting mRNA for the respective genes

| Gene | Forward primer (5' - 3') | Reverse primer (3' - 5') |
|---|---|---|
| *KCNIP1* | GAAAGACATCGCCTGGTGGTAT | GGCACTCATTTTTGAAGCCTCG |
| *KCNIP2* | CAGTTCGCTCCCTTCAGCATTA | TAGGTGCTGGAGTCTCCTTGAG |
| *KCNIP3* | ACCCTCCTTCTTGCTAAGTGGT | AGGCTGGCAACAGTTTTCTTCA |
| *KCNIP4* | TTGCCCAGTACCTTCTCAGACT | AACACCACTGGGGCATTCATTC |
| *HPRT1* | CCTGCTTCTCCTCAGCTTCAG | ACACCCTTTCCAAATCCTCAGC |
| *GAPDH* | GAAGGTGAAGGTCGGAGTCAAC | CATCGCCCCACTTGATTTTGGA |

DOI: https://doi.org/10.7554/eLife.39729.012

membranes and processed as described for the dot blot analysis using the anti-ß-actin (RRID:AB_476692), anti-KChIP3 (RRID:AB_10608850) or anti-GFP antibody (RRID:AB_390913) at a dilution of 1:5000, 1:500 and 1:1000 in 5% BSA/0.1% Tween/PBS, respectively. Membranes were washed and imaged with LI-COR Odyssey scanner (resolution = 84 µm) (LI-COR, Nebraska, USA). Quantification was performed with ImageJ (FIJI, version 2.0.0-rc-43/1.51 g) (*Schindelin et al., 2012*). The number of experiments was greater than three for each condition, and each experiment was done in triplicates.

## Counting of MUC5AC containing elements

Differentiated cells (control, KChIP3-KD or KChIP3-GFP) were grown on coverslips and to visualize intracellular MUC5AC granules, cells were washed with 1x PBS and fixed with 3% PFA/PBS for 30 min at room temperature. Cells were then washed with PBS and permeabilized for 10 min with 0.2% triton X-100 in 4% BSA/PBS. The anti-MUC5AC antibody was added to the cells at 1:5000 in 4% BSA/PBS overnight at 4°C. Next, cells were washed with PBS and incubated for 60 min at room temperature with a donkey anti-mouse Alexa Fluor 555 coupled antibody (Life Technologies), diluted at 1:1000 in 4% BSA/PBS, and DAPI (1:20000). Finally, cells were washed in PBS and mounted in FluorSave Reagent (Calbiochem, Billerica, MA). Images were acquired using an inverted Leica SP5 confocal microscope with a 63x Plan Apo NA 1.4 objective and analyzed using ImageJ (FIJI, version 2.0.0-rc-43/1.51 g) (*Schindelin et al., 2012*). For detection, the following laser lines were applied: DAPI, 405 nm; and Alexa Fluor 555, 561 nm. To determine the number and volume of MUC5AC positive elements, we used the 3D objects counter v2.0 tool from FIJI (*Bolte and Cordelières, 2006*). All images analysed were taken on the same day under the same conditions and the same Z-step (0.29 µm). The parameters used follow: a) Size filter between 100 – 37748736 voxels; b) Threshold was manually set using control cells at 31 and maintained the same for all images. DAPI signal was used to count the number of nuclei per field.

## KChIP3 and MUC5AC colocalization analysis

Differentiated HT29-18N2 (Control, KChIP3-KD, KChIP3-GFP or KChIP3-MUT) cells were grown on coverslips and to visualize MUC5AC and KChIP3 colocalization, differentiated HT29-18N2 cells were washed two times, at room temperature, with PBS for 5 min. The cells were then permeabilized by incubation in a buffer (IB) containing 20 mM HEPES pH 7.4, 110 mM KOAc (Potassium acetate), 2 mM MgOAc (Magnesium acetate) and 0.5 mM EGTA (adapted from [*Lorenz et al., 2008*]) with 0.001% digitonin for 5 min on ice, followed by washing for 7 min on ice with the same buffer without detergent. Cells were fixed in 4% paraformaldehyde for 15 min, further permeabilized for 5 min with 0.001% digitonin in IB and blocked with 4% BSA/PBS for 15 min. The anti-MUC5AC antibody was then added to the cells at 1:5000 in 4% BSA/PBS overnight at 4°C; anti-KChIP3 antibody was added to the cells at 1:500 in 4% BSA/PBS overnight at 4°C, and the anti-GFP antibody was added to the cells at 1:1000 in 4% BSA/PBS overnight at 4°C. After 24 hr, cells were washed with PBS and incubated for 1 hr at room temperature with a donkey anti-rabbit Alexa Fluor 555 (for GFP), anti-mouse Alexa Fluor 647 (for MUC5AC or KChIP3) (Life Technologies), diluted at 1:1000 in 4% BSA/PBS, and DAPI (1:20000). Finally, cells were washed in PBS and mounted in FluorSave Reagent (Calbiochem, Billerica, MA). Images were acquired using an inverted Leica SP8 confocal microscope with a 63x Plan Apo NA 1.4 objective and analysed using ImageJ ((FIJI, version 2.0.0-rc-43/1.51 g) (*Schindelin et al., 2012*). For detection of the respective fluorescence emission, the following laser lines were applied: DAPI, 405 nm; and Alexa Fluor 555, 561 nm; Alexa Fluor 647, 647 nm.

Two-channel colocalization analysis was performed using ImageJ, and the Manders' correlation coefficient was calculated using the plugin JaCop (*Bolte and Cordelières, 2006*). To determine the volume of KChIP3 positive elements, we used the 3D objects counter v2.0 tool from FIJI (*Bolte and Cordelières, 2006*). All images analysed were taken on the same day under the same conditions and the same Z-step (0.29 µm). The parameters used follow: a) Size filter between 100 – 37748736 voxels; b) Threshold was manually set at 19 for both KChIP3-GFP and KChIP3-MUT cells and maintained the same for all images. DAPI signal was used to count the number of nuclei per field.

## Measurement of intracellular [Ca$^{2+}$]

Differentiated HT29-18N2 cells (Control, KChIP3-GFP and KChIP3-KD) were plated on glass cover-slips, loaded with 5 µM of Fura-2AM for 30 min at room temperature, washed and bathed in an iso-tonic solution containing: 140 mM NaCl, 2.5 mM KCl, 1.2 mM CaCl$_2$, 0.5 mM MgCl$_2$, 5 mM glucose, 10 mM HEPES (305 mosmol/l, pH 7.4 adjusted with Tris). Cytosolic Ca$^{2+}$ levels were measured in the absence of stimuli (for all conditions) for 15 min and 100 µM ATP added to the bath solution as indi-cated in the figure legend. All experiments were carried out at room temperature as described pre-viously (*Fernandes et al., 2008*). AquaCosmos software (Hamamatsu Photonics) was used for capturing the fluorescence ratio at 505 nm obtained post-excitation at 340 and 380 nm. Images were computed every 5 s. Measurements were processed using SigmaPlot 10 software.

## *Kcnip3*$^{-/-}$ mice

*Kcnip3*$^{-/-}$ mice were generated on C57BL/6 strain (*Cheng et al., 2002*), and WT C57BL/6 mice were used as a control for this study (both sets of animals were obtained from Dr. Naranjo's Lab). In order to evaluate the mucus layer under basal situation on healthy mice, 12 week old male mice were used. Specifically, six animals per genotype (wild type and *Kcnip3*$^{-/-}$ mice) were evaluated.

## Histological study

Mice were sacrificed by CO$_2$ inhalation and necropsy was performed to obtain the colon. Half of each intestinal segment were cleaned of faeces prior to its fixation with 10% NBF and the other half were maintained with the faecal pellets (when present) for its fixation with Carnoy's fixative to better preserve the mucin layer. All samples were fixed overnight by incubation at 4°C and cut in 4 µm sections.

## PAS and PAS-AB staining

Colon mice sections were stained with hematoxylin and eosin (H/E), periodic acid–Schiff (PAS) and Alcian Blue plus periodic acid–Schiff (PAS-AB) for histological analysis. PAS was used to stains neu-tral, acid-simple non-sulfated and acid-complex sulfated mucins (mucins are stained in purple/ magenta). For the PAS-AB staining, it first stains the acidic mucins with Alcian blue; those remaining acidic mucins that are also PAS positive will be chemically blocked and will not react further. Those neutral mucins that are solely PAS positive will subsequently be demonstrated in a contrasting man-ner. Where mixtures occur, the resultant colour will depend upon the dominant moiety. For PAS-AB staining, acidic mucins are stained in blue, neutral mucins in magenta and mixtures in blue/purple.

Full images of PAS and PAS-AB stained sections were acquired by a NanoZoomer-2.0 HT C9600 scanner (Hamamatsu) at 20X magnification, in which one pixel corresponds to 0.46 µm.

## Measurement of the mucus layer thickness

The thickness of mucus layer was measured in medium and distal colonic tissue sections using the ruler tool of the NDP view + 2.50.19 software (Hamamatsu) in both PAS and PAS-AB stained sec-tions. 20 different measures were performed in two different tissue sections per stain in both medium and distal colon. Mucus layer thickness has been evaluated in all the samples that presented faecal pellets.

## Statistical analysis

All data are means ± SEM. In all cases a D'Agostino– Pearson omnibus normality test was performed before any hypothesis contrast test. Statistical analysis and graphics were performed using Graph-Pad Prism 6 (RRID:SCR_002798) or SigmaPlot 10 (RRID:SCR_003210) software. For data that fol-lowed normal distributions, we applied either Student's t test or one-way analysis of variance (ANOVA) followed by Tukey's post hoc test. For data that did not fit a normal distribution, we used Mann–Whitney's unpaired t test and nonparametric ANOVA (Kruskal–Wallis) followed by Dunn's post hoc test. Criteria for a significant statistical difference were: *p<0.05; **p<0.01.

## Acknowledgements

We thank all members of the Malhotra Lab, especially Ishier Raote, for valuable discussions. Cell sorting experiments were carried out by the joint CRG/UPF FACS Unit at Parc de Recerca Biomèdica de Barcelona (PRBB). Fluorescence microscopy was performed at the Advanced Light Microscopy Unit at the CRG, Barcelona. V.Malhotra is an Institució Catalana de Recerca i Estudis Avançats professor at the Centre for Genomic Regulation. This work was funded by grants from the Spanish Ministry of Economy and Competitiveness (BFU2013-44188-P to VM, SAF2015-69762R to MAV and SAF2017-89554-R to JRN) and FEDER Funds. We acknowledge support of the Spanish Ministry of Economy and Competitiveness, through the Programmes "Centro de Excelencia Severo Ochoa 2013- 2017" (SEV-2012-0208 & SEV-2013-0347) and Maria de Maeztu Units of Excellence in R&D (MDM-2015- 0502). This work reflects only the authors' views, and the EU Community is not liable for any use that may be made of the information contained therein.

## Additional information

### Competing interests

Vivek Malhotra: Senior Editor at eLife. The other authors declare that no competing interests exist.

### Funding

| Funder | Grant reference number | Author |
| --- | --- | --- |
| Ministerio de Economía y Competitividad | BFU2013-44188-P | Vivek Malhotra |
| Ministerio de Economía y Competitividad | SAF2015-69762R | Miguel A Valverde |
| Ministerio de Economía y Competitividad | SAF2017-89554-R | José Ramón Naranjo |
| Centro de Excelencia Severo Ochoa 2013-2017 | SEV-2012-0208 | Vivek Malhotra |
| Centro de Excelencia Severo Ochoa 2013-2017 | SEV-2013-0347 | Vivek Malhotra |
| Maria de Maeztu Units of Excellence | MDM-2015- 0502 | Vivek Malhotra |

The funders had no role in study design, data collection and interpretation, or the decision to submit the work for publication.

### Author contributions

Gerard Cantero-Recasens, Conceptualization, Data curation, Formal analysis, Supervision, Validation, Investigation, Visualization, Methodology, Writing—original draft, Writing—review and editing; Cristian M Butnaru, Formal analysis, Validation, Investigation, Visualization, Methodology, Writing—review and editing; Miguel A Valverde, Formal analysis, Funding acquisition, Methodology, Writing—review and editing; José R Naranjo, Resources, Writing—review and editing; Nathalie Brouwers, Investigation, Methodology, Writing—review and editing; Vivek Malhotra, Conceptualization, Formal analysis, Supervision, Funding acquisition, Writing—original draft, Project administration, Writing—review and editing

### Author ORCIDs

Gerard Cantero-Recasens http://orcid.org/0000-0001-6452-782X
Nathalie Brouwers http://orcid.org/0000-0002-9808-9394
Vivek Malhotra http://orcid.org/0000-0001-6198-7943

### Ethics

Animal experimentation: Animal care was conducted in accordance with standard ethical guidelines (European Communities Directive 86/609 EEC; National Institutes of Health 1995). The CNB-CSIC and Community of Madrid ethical committees approved experiments with mice (PROEX 28/05).

### Decision letter and Author response

Decision letter https://doi.org/10.7554/eLife.39729.015
Author response https://doi.org/10.7554/eLife.39729.016

## Additional files

### Supplementary files

• Transparent reporting form
DOI: https://doi.org/10.7554/eLife.39729.013

### Data availability

All data generated or analysed during this study are included in the manuscript and supporting files.

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
