## [Decision Letter]

[**Editorial note:** This article has been through an editorial process in which the authors decide how to respond to the issues raised during peer review. The Reviewing Editor's assessment is that all the issues have been addressed.]

Thank you for submitting your article "KChIP3 coupled to Ca^2+^ oscillations exerts a tonic brake on baseline mucin release in the colon" for consideration by *eLife*. Your article has been reviewed by three peer reviewers, including Suzanne R Pfeffer as the Reviewing Editor and Reviewer #1, and the evaluation has been overseen by Richard Aldrich as the Senior Editor. The following individual involved in review of your submission has agreed to reveal their identity: Burton Dickey (Reviewer #2). Reviewer #3 remains anonymous.

The Reviewing Editor has highlighted the concerns that require revision and/or responses, and we have included the separate reviews below for your consideration. If you have any questions, please do not hesitate to contact us.

Cantero-Recasens and colleagues show that knocking down of KChIP3 leads to mucin hypersecretion, indicating a previously unanticipated, inhibitory role of KChIP3 in baseline secretion. In support of this notion, the authors showed that overexpression of KChIP3 WT or Ca^2+^ Mut inhibits baseline mucin secretion. Furthermore, the authors found that both KChIP3 WT and Ca^2+^ Mut associate with MUC5AC granules. Increases in Ca^2+^-oscillations by EGTA treatments significantly reduced the granule localization of KChIP3 WT. In contrast, KChIP3 Ca^2+^ Mut showed no changes upon EGTA treatments. These findings have led the authors to conclude that KChIP3 is likely to be the high-affinity Ca^2+^ sensor for baseline mucin secretion, and they further propose that KChIP3 marks mature and primed mucin granules, and functions as a Ca^2+^-dependent brake for baseline mucin secretion. As noted in the reviews below, the reviewers agree that the findings are of broad interest and suitable novelty for presentation in *eLife*. However, they request textual clarification of a number of issues, and look forward to your suggestions how to address with their various concerns described below. Please note that all of the comments are offered to enhance the clarity of the presentation and the interpretability of the experiments shown.

Separate reviews (please respond to each point):

*Reviewer #1:*

This is an interesting manuscript that describes the role of a novel calcium regulator that influences constitutive mucin secretion in relation to ER calcium oscillations via the ryanodine receptor. That this protein is associated with the granules and puts a brake on the process is an important contribution and my comments are offered to improve the clarity of the story.

1) Please delete "The description of our findings follows" (this is obvious).

2) Subsection “KChIP3 is required for baseline mucin secretion in colonic goblet cells”, end of first paragraph. Please provide a value for how much the protein was overexpressed.

3) Subsection “KChIP3 localizes to a pool of mucin secretory granules”, first paragraph. This reviewer does not understand what the punctae and 10-15% reflects in relation to the images shown. Please clarify.

4) Subsection “KChIP3^-/-^ mice show increased colonic mucus layer under basal conditions”, second paragraph. Please add the word, significantly and show NS instead of a p value.

5) Subsection “KChIP3^-/-^ mice show increased colonic mucus layer under basal conditions”, second paragraph. Is there any way that the authors can use a more specific stain than PAS or PAS-AB? Please clarify how specific this is and if needed, justify its use; it would be so much better to use an anti-mucin antibody. Or?

6) Discussion should do a better job of summarizing other people's work on calsenilin and ryanodine receptor and how the soluble protein may function or how the protein could act in relation to ER.

Figure 1 – blue versus black dots are hard to see; Figure 1F – it is impossible to understand how you handled what looks to be a green smoosh and counted objects in control cells. This really needs to be explained more clearly. Did you determine total green pixels? How did you threshold this? Please clarify.

Figure 1—figure supplement 2C, D. y axis should include the word, protein.

More information is needed regarding how Figure 1 images were quantified.

*Reviewer #2:*

The major strength of the work of Cantero-Recasens et al. is the discovery of a possible novel regulator of mucin secretion. This is important because the discovery of new participants in exocytic events at this point in time is challenging in view of the extensive research that has been done on molecular mechanisms of exocytosis during the past three decades. As a result, most current research focuses on refinement of understanding of the roles of known participants. However, there are multiple weaknesses in the way the work is interpreted and presented, and specific suggestions are offered here to present the story with greater scholarly precision and clarity.

1) A critical issue related to the experiments requires clarification. By measuring MUC5AC in cellular supernatants, the authors detect a 2.5-fold increase in baseline secretion (subsection “KChIP3 is required for baseline mucin secretion in colonic goblet cells”, first paragraph), and other changes up and down in subsequent assays. However, there should be no measurable change in baseline (steady state) secreted mucins unless there is a secondary change in mucin gene expression, which the authors rule out. This issue is addressed in detail in Ren et al., 2015, wherein alterations in baseline secretion were analyzed in airway mucous cells in SNAP23 heterozygous knockout mice. In that paper, heterozygous loss of SNAP23 results in spontaneous intracellular mucin accumulation without inflammation or increased mucin gene expression. The increased intracellular pool brings the rate of mucin release back to wildtype, which can be conceptualized in a hydrodynamic model illustrated in Figure 7 of that paper. In fact, if one thinks about it, the steady state of baseline secretion necessarily reflects the rate of mucin synthesis or cells would become engorged until they lysed or they reduced mucin synthesis if there were a reduction in baseline secretion, or the converse if there were an increase. These issues were further explored in Zhu et al., 2015. The same reasoning calls the in vivo results shown in Figure 4, into question, and in this regard it is concerning that there is not a clear depletion of intracellular mucin stores in the intestine. Intracellular mucin depletion should be the major phenotype of an increase in baseline secretion (opposite to the mucin accumulation phenotype observed in Zhu et al., 2008, and Ren et al., 2015, in baseline secretion defective mice). The authors should carefully measure intracellular mucin in WT and mutant mice, and fully address this issue.

2) In Figure 1E, the calculation of ATP-dependent secretion from the data in D as the difference between normalized baseline and stimulated secretion for each condition does not seem justified. In Figure 1—figure supplement 1B, the full blot should be shown so the relative level of the overexpressed KChIP3-GFP fusion can be compared to endogenous KChIP3 (also, it's cumbersome for the reader to label the top row a-DREAM, which is a synonym for KChIP3, even if that is the preferred name by the vendor). In Figure 1—figure supplement 1C, is overexpression of KChIP3 really 500-fold? The figure legends are very long.

3) Regarding calcium oscillations, are the authors certain that it is the oscillations themselves that are responsible for regulating mucin secretion versus mean cytoplasmic calcium concentrations?

4) First sentence of the Abstract "Regulated secretion by specialized goblet cells in agonist-dependent (stimulated) and -independent (baseline) manners.…".

If secretion is "regulated", then it should not occur in an agonist-independent manner, and it probably doesn't. The authors are correct when they assert that even baseline secretion is regulated, at least in airway cells, as demonstrated several ways. First, baseline secretion depends on the exocytic sensor of second messengers Munc13-2 (Zhu et al., 2008). Second, it depends on the second messenger calcium (Rossi, Sears and Davis, 2004). Third, it depends on the P2Y2 purinergic receptor (references in Davis and Dickey, 2008). Fourth, it depends on shear stress, probably mediated by ATP released mostly by neighboring ciliated cells during respiration (Zhu et al., 2015; Tarran R, Annu Rev Physiol, 2006). A simple way to frame this is in terms of low and high levels of agonists (or second messengers) rather than agonist-independent and agonist-dependent. Presumably there is a constitutive secretory pathway that exists in parallel to the baseline and stimulated regulated pathways, and this reviewer is aware of some presumptive molecular components of the constitutive pathway in unpublished work. (Oddly, the authors seem to have awareness of these issues because, in their prior work, Mitrovic et al., *eLife*2013, the authors correctly noted that "fusion of MUC5AC-containing granules with the plasma membrane requires an external signal."). Please rewrite the Abstract and Introduction with care to make these points more clearly.

5) In the second to last sentence of the Abstract, the statement "KChIP3 emerges as the long-sought high-affinity Ca^2++^ sensor that regulates baseline mucin secretion" is likely to confuse those in the field because KChIP3 appears to be purely a negative regulator. Syt2 was identified as the low-affinity calcium sensor in stimulated secretion in airway cells, as correctly noted by the authors. However, Syt2 is a positive regulator of stimulated secretion in both airway cells and neurons, because its deletion in airway mucous cells results in a severe reduction of ATP-stimulated mucin release (Tuvim et al., 2009) and its deletion in neurons results in a severe reduction of evoked release in neurons (Pang ZP, J Neurosci, 2006). Besides its positive activity, Syt2 also has a negative regulatory activity because its deletion in neurons results in increased asynchronous and spontaneous synaptic vesicle release, but whether it has a negative activity in mucous cells is not known. In any case, what has long been sought is the high-affinity calcium sensor that positively regulates baseline mucin secretion. While Cantero-Recasens et al. have apparently discovered an interesting regulatory pathway, they need to explain these differences with care as not to confuse the reader.

6) In the Discussion, it is stated, "in a nutshell, all that is known (about baseline mucin secretion) is that it is independent of Syt2 and is sensitive to mechanical stress." However, we also know that SNAP23 mediates both baseline and stimulated secretion (Ren et al., 2015), and that Munc13-2 regulates both processes (Zhu et al., 2008). Please cite these papers.

7) The authors go back and forth between airway and intestinal goblet cells as if they have identical mechanisms mediating and regulating secretion when in fact they don't. To begin, airway secretion, both baseline and stimulated, depends upon shear forces (airflow in vivo, liquid flow in vitro) mediated by ATP release from neighboring cells, whereas in the intestine, there is an adherent layer of MUC2. Another example is that in the intestine, sentinel goblet cells at the entrance to crypts respond to TLR signaling with massive secretion (Birchenough GM, Science, 2016), whereas no such phenomenon has been detected in the airway. Therefore, the authors should better clarify what is known in the airway, what is known in the intestine, and what, exactly, they are modeling. There is space to do this.

8) Minor issues: Subsection “KChIP3^-/-^ mice show increased colonic mucus layer under basal conditions”, second paragraph, it is stated that an increase in the mucus layer was detected in the medial colon of four out of five knockout mice; was the fifth mouse excluded from the data analysis? In the next paragraph, the comment in Results that "our data showing an increased mucus layer in KO mice fit nicely with data obtained in vitro" is more appropriate for Discussion.

9) In the Introduction, it is confusing to state, "our genome-wide screen identified new proteins that control extracellular calcium entry into cells.… also identified was a high-affinity calcium-binding protein, KChIP3". This makes it sound as if KChIP3 regulates calcium entry, when it is subsequently demonstrated that it does not. It should also be pointed out that in airway mucous cells over short time frames, extracellular calcium does not contribute to the level of cytoplasmic calcium (referenced in Tuvim et al., 2009, and Davis and Dickey, 2008).

10) There is room for improvement in phrasing in nearly every paragraph, with some being important (see below) and some trivial, for example, starting with the Abstract, granules are not released, rather their contents are released, and this could simply be fixed as "allowing for their exocytosis".

In the Introduction, they talk about "a Ca^2+^-mediated reaction to a gel-like form". This is likely to be very confusing to readers because the manuscript revolves around cytoplasmic calcium sensing, while the effect of calcium on mucus gel-like properties is an entirely different subject that does not come up elsewhere in the manuscript. It is true that calcium binds mucins within secretory granules, and then is exchanged for sodium in the extracellular space after mucin secretion, but it would be simplest to just drop this fact from the Introduction.

In the third paragraph of the Introduction, it would be helpful to add a few words "identified new proteins, including TRPM5", to orient the reader to the prior work. “entry into the cells” is better as "entry into cells", and “vicinity of exocytic machinery” as "vicinity of the exocytic machinery".

The choice of KChIP3-WT as the designation of the transgenic overexpressing line is confusing, because WT can refer either to WT sequence or WT level of expression; it would be better to designate it as OE for overexpressing or GFP since that is now inserted.

Subsection “KChIP3 is required for baseline mucin secretion in colonic goblet cells”, second paragraph, it would be helpful if the authors provided a short description of circumstances when MUC5AC is expressed in colonic epithelium, such as parasite infection or ulcerative colitis, as they did in Mitrovic.

The apparent rigor of statistical analysis is low.

*Reviewer #3:*

Cantero-Recasens and colleagues presented an interesting study on the impact of KChIP3 on mucin secretion. They showed that knocking down of KChIP3 lead to mucin hypersecretion, indicating an inhibitory role of KChIP3 in baseline secretion. In support of this notion, the authors showed that overexpression of KChIP3 WT or Ca^2+^ Mut inhibits baseline mucin secretion. Furthermore, the authors found that both KChIP3 WT and Ca^2+^ Mut associate with MUC5AC granules. Increases in Ca^2+^-oscillations by EGTA treatments significantly reduced the granule localization of KChIP3 WT. In contrast, KChIP3 Ca^2+^ Mut showed no changes upon EGTA treatments. These findings have led the authors to conclude that KChIP3 is likely to be the high-affinity Ca^2+^ sensor for baseline mucin secretion, and they further proposed that KChIP3 marks mature and primed mucin granules, and functions as a Ca^2+^-dependent brake for baseline mucin secretion. In general, the experiments in this study were carefully designed and well executed. The inhibitory activity of KChIP3 on baseline mucin secretion is nicely shown. A potential weakness of this study is the link between Ca^2+^ and the dissociation of KChIP3 from granules.

KChIP3 granule localization is interesting, and this appears to be the main evidence for the proposed mechanism underlying KChIP3 inhibition. However, it is unclear to this reviewer what is the mechanistic nature behind the reduction of KChIP3 localization upon the EGTA treatment. For example, KChIP3 is a multifunctional protein that binds proteins and DNA, and it regulates gene expression in a Ca^2+^-dependent way. Can the authors exclude the possibility that the changes of KChIP3-granule localization are due to alterations of protein/membrane components of granules on a longer timescale? Also, have the authors tried to block granule exocytosis by knocking down SNAREs (e.g., VAMP8)? Does an impairment of granule exocytosis inhibit KChIP3 dissociation? At this stage, further evidence to support the model in Figure 5 would be helpful.

In Figure 3, the authors presented data on KChIP3 staining using permeabilized and washed cells. While it is clear that some KChIP3 proteins are co-localized with granules, the degree of KChIP3 granular/cytosolic localization is unclear. What percentage of total KChIP3 is on granules? Without washing, could the authors observe KChIP3 WT and Ca^2+^ Mut on docked granules (e.g., by TIRF)? Does the EGTA treatment change KChIP3 localization in intact cells? It would be helpful to know whether the EGTA treatment alters KChIP3 distribution by triggering complete dissociation or by changing the tightness of KChIP3-granule interactions.

[Editors' note: further revisions were requested prior to acceptance, as described below.]

Thank you for submitting your article "KChIP3 coupled to Ca^2+^ oscillations exerts a tonic brake on baseline mucin release in the colon" for consideration by *eLife*.

The authors have very thoughtfully responded to the detailed reviewer comments. However, this monitoring editor would strongly encourage the authors to incorporate most of their responses directly into the manuscript text in the Results section. If a reviewer expert has a question or is confused, other readers will too, and discussion of the issues in the paper will greatly improve the manuscript. For example, please state in the legend, "Shown are Z-stack projections but quantification utilized data from individual stacks using 3D analysis software." A reader should not need to go to the methods to figure this out. With this suggestion, the paper will represent an important contribution to the field overall.

---

## [Author Response]

Reviewer #1:

This is an interesting manuscript that describes the role of a novel calcium regulator that influences constitutive mucin secretion in relation to ER calcium oscillations via the ryanodine receptor. That this protein is associated with the granules and puts a brake on the process is an important contribution and my comments are offered to improve the clarity of the story.1) Please delete "The description of our findings follows" (this is obvious).

This statement is now removed from the text.

2) Subsection “KChIP3 is required for baseline mucin secretion in colonic goblet cells”, end of first paragraph. Please provide a value for how much the protein was overexpressed.

For this experiment, we used the KChIP3-WT cell line (now renamed to KChIP3-GFP), and expression of KChIP3 was checked by qPCR and confirmed by western blot (WB) (Figure 1—figure supplement 1B and C). Available commercial antibodies against KChIP3 do not detect endogenous levels of KChIP3 by western blots. We can, therefore, only provide a value of how much KChIP3 is overexpressed at the RNA level but not at the protein level compared to endogenous levels. To avoid any confusion, we have changed the sentence as follows:

“Conversely, overexpression of KChIP3 (using KChIP3-GFP cells) produced a 30% reduction in baseline MUC5AC secretion (Figure 1D), without affecting ATP-dependent MUC5AC secretion (Figure 1E).”

3) Subsection “KChIP3 localizes to a pool of mucin secretory granules”, first paragraph. This reviewer does not understand what the punctae and 10-15% reflects in relation to the images shown. Please clarify.

In order to study the localization of KChIP3 in HT29 differentiated cells, we used the KChIP3-GFP cell line and detected, by confocal immunofluorescence, the signal of both KChIP3 and MUC5AC. Next, we studied the colocalization between these two proteins using FIJI software for all z-stacks. Our data show that 10-15% of total MUC5AC containing granules detected by immunofluorescence (IF) contain KChIP3-GFP. However, the images shown in Figure 3 and Figure 3—figure supplement 4 represent an apical single plane and, therefore, not all mucin granules are visible.

4) Subsection “KChIP3^-/-^ mice show increased colonic mucus layer under basal conditions”, second paragraph. Please add the word, significantly and show NS instead of a p value.

We have made the changes as suggested.

5) Subsection “KChIP3^-/-^ mice show increased colonic mucus layer under basal conditions”, second paragraph. Is there any way that the authors can use a more specific stain than PAS or PAS-AB? Please clarify how specific this is and if needed, justify its use; it would be so much better to use an anti-mucin antibody. Or?

PAS and PAS-AB are commonly used to stain both acidic and neutral mucins. Although it is true that PAS technique is also used for the detection of glycogen and other glycoproteins, the combination of both PAS and Alcian-Blue is, by far, the most sensitive method to detect mucins, since all mucins are detected regardless of their charge. Since our data (Figure 1 and Figure 1—figure supplement 2) show that KChIP3 has similar effect on both MUC2 and MUC5AC export, staining the sections with a specific mucin antibody will not provide any valuable information. Another key issue is that due to hyperglycosylation of mucins (80% sugars and 20% polypeptide), their detection by immunohistochemistry is technically very challenging. This would be a huge undertaking without the possibility of a convincing result better than the data presented.

6) Discussion should do a better job of summarizing other people's work on calsenilin and ryanodine receptor and how the soluble protein may function or how the protein could act in relation to ER.

We have now included the following paragraph in the Discussion:

“In neurons, KChIP3 is shown to alter ER calcium content (Lilliehook, 2002) and RYR-mediated Ca^2+^-induced Ca^2+^ release (CICR) through direct interaction with RYR receptors (Grillo, 2018). Our data show that in HT29 goblet cells, however, modulating KChIP3 levels does not alter intracellular Ca^2+^ homeostasis; but, instead, its localization to granules was regulated by Ca^2+^ oscillations. Importantly, since ER is in close proximity to mucin granules in goblet cells (Tuvim, 2009), it is possible that there is a direct interaction between RYR and KChIP3, which could efficiently influence KChIP3 activity and localization by ER-based Ca^2+^ oscillations, for example, by promoting KChIP3 detachment from the granules and coupling ER Ca^2+^ release to baseline mucin secretion.”

Figure 1 – blue versus black dots are hard to see.

We have changed the shape and color of blue dots.

Figure 1F – it is impossible to understand how you handled what looks to be a green smoosh and counted objects in control cells. This really needs to be explained more clearly. Did you determine total green pixels? How did you threshold this? Please clarify.

The images are z-projections, but we quantified all the z-stacks using FIJI software. We have now included a new paragraph in the Materials and methods to explain this procedure as shown below.

“Counting of MUC5AC containing elements

Differentiated cells (control, KChIP3-KD or KChIP3-GFP) were grown on coverslips and to visualize intracellular MUC5AC granules, cells were washed with 1x PBS and fixed with 3% PFA / PBS for 30 min at room temperature. Cells were then washed with PBS and permeabilized for 10 min with 0.2% triton X-100 in 4% BSA / PBS. The anti-MUC5AC antibody was added to the cells at 1:5000 in 4% BSA / PBS overnight at 4ºC. Next, cells were washed with PBS and incubated for 60 min at room temperature with a donkey anti-mouse Alexa Fluor 555 coupled antibody (Life Technologies), diluted at 1:1000 in 4% BSA / PBS, and DAPI (1:20000). Finally, cells were washed in PBS and mounted in FluorSave Reagent (Calbiochem, Billerica, MA). Images were acquired using an inverted Leica SP5 confocal microscope with a 63x Plan Apo NA 1.4 objective and analyzed using ImageJ (FIJI, version 2.0.0-rc-43/1.51g) (Schindelin et al., 2012). For detection, the following laser lines were applied: DAPI, 405 nm; and Alexa Fluor 555, 561 nm. To determine the number and volume of MUC5AC positive elements, we used the 3D objects counter v2.0 tool from FIJI (Bolte and Cordelières, 2006). All images analysed were taken on the same day under the same conditions and the same Z-step (0.29 µm). The parameters used follow: a) Size filter between 100-37748736 voxels; b) Threshold was manually set using control cells at 31 and maintained the same for all images. DAPI signal was used to count the number of nuclei per field.”

Figure 1—figure supplement 2C, D. y axis should include the word, protein.

We have included the word “protein” on the y-axis.

More information is needed regarding how Figure 1 images were quantified.

As stated before, now we include a new paragraph in the Materials and methods explaining how images were quantified.

“*Counting of MUC5AC containing elements*

Differentiated cells (control, KChIP3-KD or KChIP3-GFP) were grown on coverslips and to visualize intracellular MUC5AC granules, cells were washed with 1x PBS and fixed with 3% PFA / PBS for 30 min at room temperature. Cells were then washed with PBS and permeabilized for 10 min with 0.2% triton X-100 in 4% BSA / PBS. The anti-MUC5AC antibody was added to the cells at 1:5000 in 4% BSA / PBS overnight at 4ºC. Next, cells were washed with PBS and incubated for 60 min at room temperature with a donkey anti-mouse Alexa Fluor 555 coupled antibody (Life Technologies), diluted at 1:1000 in 4% BSA / PBS, and DAPI (1:20000). Finally, cells were washed in PBS and mounted in FluorSave Reagent (Calbiochem, Billerica, MA). Images were acquired using an inverted Leica SP5 confocal microscope with a 63x Plan Apo NA 1.4 objective and analyzed using ImageJ (FIJI, version 2.0.0-rc-43/1.51g) (Schindelin et al., 2012). For detection, the following laser lines were applied: DAPI, 405 nm; and Alexa Fluor 555, 561 nm. To determine the number and volume of MUC5AC positive elements, we used the 3D objects counter v2.0 tool from FIJI (Bolte and Cordelières, 2006). All images analysed were taken on the same day under the same conditions and the same Z-step (0.29 µm). The parameters used follow: a) Size filter between 100-37748736 voxels; b) Threshold was manually set using control cells at 31 and maintained the same for all images. DAPI signal was used to count the number of nuclei per field.”

Reviewer #2:

The major strength of the work of Cantero-Recasens et al. is the discovery of a possible novel regulator of mucin secretion. This is important because the discovery of new participants in exocytic events at this point in time is challenging in view of the extensive research that has been done on molecular mechanisms of exocytosis during the past three decades. As a result, most current research focuses on refinement of understanding of the roles of known participants. However, there are multiple weaknesses in the way the work is interpreted and presented, and specific suggestions are offered here to present the story with greater scholarly precision and clarity.1) A critical issue related to the experiments requires clarification. By measuring MUC5AC in cellular supernatants, the authors detect a 2.5-fold increase in baseline secretion (subsection “KChIP3 is required for baseline mucin secretion in colonic goblet cells”, first paragraph), and other changes up and down in subsequent assays. However, there should be no measurable change in baseline (steady state) secreted mucins unless there is a secondary change in mucin gene expression, which the authors rule out. This issue is addressed in detail in Ren et al., 2015, wherein alterations in baseline secretion were analyzed in airway mucous cells in SNAP23 heterozygous knockout mice. In that paper, heterozygous loss of SNAP23 results in spontaneous intracellular mucin accumulation without inflammation or increased mucin gene expression. The increased intracellular pool brings the rate of mucin release back to wildtype, which can be conceptualized in a hydrodynamic model illustrated in Figure 7 of that paper. In fact, if one thinks about it, the steady state of baseline secretion necessarily reflects the rate of mucin synthesis or cells would become engorged until they lysed or they reduced mucin synthesis if there were a reduction in baseline secretion, or the converse if there were an increase. These issues were further explored in Zhu et al., 2015. The same reasoning calls the in vivo results shown in Figure 4, into question, and in this regard it is concerning that there is not a clear depletion of intracellular mucin stores in the intestine. Intracellular mucin depletion should be the major phenotype of an increase in baseline secretion (opposite to the mucin accumulation phenotype observed in Zhu et al., 2008, and Ren et al., 2015, in baseline secretion defective mice). The authors should carefully measure intracellular mucin in WT and mutant mice, and fully address this issue.

As shown in Figure 1F, knock-down of KChIP3 in HT29 cells results in a reduction of intracellular MUC5AC staining, in accordance with the phenotype of an increase in baseline secretion that should produce a depletion of intracellular mucins. In the colon of the KChIP3 KO mice we observed a clear increase in the colonic mucus layer thickness. However, it was not possible to measure intracellular mucin staining. The only possibility was to measure mucin staining of the colonic epithelia, which includes goblet cells, the epithelium lining and the mucins secreted in the colonic crypt tube using the Positive Pixel Count Algorithm. Briefly, a region of interest (ROI) of colonic epithelia was selected for each image and pixels segmented in positive (PAS or PAS-AB positive pixels) or negative (rest of the pixels), and then analyzed as the percentage of positive pixels (number of positive pixels*100/(positive pixels+negative pixels)). However, since it was not possible to distinguish between the mucins secreted into the colonic crypt tube and the intracellular mucins, there was no observable difference between the WT and KChIP3 KO mice.

Therefore, the only way to quantify the differences in intracellular mucins between the KO and the WT mice would be to isolate the colonic cells, harvest and visualize by immunofluorescence and/or dot blotting with anti-mucin antibodies (after extensive washing to remove all secreted mucins). However, this is technically challenging and likely to introduce artifacts: the process of isolation might trigger mucin release and yield a false report.

Importantly, we show that in the tissue culture cells, there are fewer intracellular granules in KChIP3 depleted cells, so it is very likely that this event is reproduced in the mouse KO, but at this stage beyond our abilities to test for the reasons stated above.

2) In Figure 1E, the calculation of ATP-dependent secretion from the data in D as the difference between normalized baseline and stimulated secretion for each condition does not seem justified.

We decided to use the difference between baseline and stimulated secretion to calculate the real contribution of KChIP3 to ATP-mediated mucin secretion since in our experimental approach stimulated secretion also includes baseline secretion. In the case of KChIP3-KD cells there is an apparent increase of stimulated mucin secretion, which is due to an increase in baseline secretion. We therefore calculated ATP-dependent mucin secretion as the difference between stimulated and unstimulated secretion.

In Figure 1—figure supplement 1B, the full blot should be shown so the relative level of the overexpressed KChIP3-GFP fusion can be compared to endogenous KChIP3 (also, it's cumbersome for the reader to label the top row a-DREAM, which is a synonym for KChIP3, even if that is the preferred name by the vendor).

Using available commercial antibodies against KChIP3, we have been unable detect endogenous KChIP3 by WB. We therefore show the band corresponding to the overexpressed KChIP3 (using anti-KChIP3 and anti-GFP antibodies) as a means to represent the KChIP3 levels. We have changed the label from a-DREAM to a-KChIP3.

In Figure 1—figure supplement 1C, is overexpression of KChIP3 really 500-fold?

In Figure 1—figure supplement 1C, there is a 500-fold increase in KChIP3 RNA levels (normalized by GAPDH levels) in KChIP3 overexpressing cells compared to control cells. We have changed expression levels to relative RNA levels on the y-axis.

The figure legends are very long.

We have shortened the figure legends as shown below.

“*Figure 1. KChIP3 levels regulate baseline MUC5AC secretion*

A) KChIP3 RNA levels from undifferentiated (UD) and differentiated HT29-18N2 cells normalized by GAPDH values. B) Control (black circles) and KChIP3 stable knockdown cells (KChIP3-KD) (blue squares) were differentiated and incubated for 30 minutes at 37ºC in the absence or presence of 100 µM ATP. Secreted MUC5AC was collected and dot blotted with an anti-MUC5AC antibody. Data were normalized to actin levels. The y-axis represents normalized values relative to the values of untreated control cells. C) ATP-dependent MUC5AC secretion was calculated from the data in (B) as the difference between normalized baseline secretion and stimulated secretion for each condition. D) Secreted MUC5AC from differentiated control (black circles) and KChIP3 overexpressing cells (KChIP3-GFP) (red circles) in the absence or presence of 100 µM ATP. E) ATP-dependent MUC5AC secretion calculated from the data in (D) for each condition. F) Immunofluorescence of control, KChIP3-KD and KChIP3-GFP differentiated HT29-18N2 cells with anti-MUC5AC antibody (green) and DAPI (red). G) The number of MUC5AC granules for control (black circles), KChIP3-KD (blue squares) and KChIP3-GFP (red circles) cells was quantified from immunofluorescence images using FIJI software. The y-axis represents the number of 3-D objects detected by the software divided by the number of cells in each field. Scale bar = 5 µM. H) Volume of control (black), KChIP3-KD (blue) and KChIP3-GFP (red) MUC5AC granules was calculated from immunofluorescence images using FIJI software. The y-axis represents the volume of the granules in µm^3^. Abbreviations: UD: Undifferentiated HT29-18N2 cells, DF: Differentiated HT29-18N2 cells. * p< 0.05, ** p<0.01.”

“*Figure 2. Ca^2+^ oscillations in goblet cells control KChIP3 function*

A) Time course of Ca^2+^ responses (normalized FURA-2AM ratio) obtained in differentiated HT29-18N2 cells in resting conditions exposed to different extracellular buffers: 1.2 mM CaCl_2_ (left), 0.5 mM EGTA (center), or 10 µM dandrolene (right) (n=30, inset shows a recording obtained from a single cell under each condition). B) Percentage of cells oscillating in each condition during 10 minutes. Average values ± SEM are plotted as scatter plot with bar graph (N>3) (black dots: 1.2 mM CaCl_2_, blue dots: EGTA, red dots: dandrolene). C) Secreted MUC5AC collected from differentiated HT29 cells that were incubated for 30 min at 37ºC with different buffers: 1.2 mM CaCl_2_ (black dots), 0.5 mM EGTA (blue dots) or 10 µM dandrolene (red dots). The y-axis represents relative values with respect to the values of control cells. Average values ± SEM are plotted as scatter plot with bar graph (N>3). D) Secreted MUC5AC from differentiated control (black circles) and KChIP3 stable knockdown cells (KChIP3-KD) (blue squares) collected after 30 minutes incubation at 37ºC in the in the presence (1.2 mM CaCl_2_) or absence (0.5 mM EGTA) of extracellular Ca^2+^. Data were normalized to intracellular actin levels. The y-axis represents normalized values relative to the values of untreated control cells. E) Secreted MUC5AC from control (black circles) and KChIP3 stable knockdown cells (KChIP3-KD) (blue squares) that were incubated for 30 minutes at 37ºC with vehicle or 10 µM dandrolene (DAND) in the presence of extracellular Ca^2+^. Data were normalized to intracellular actin levels. The y-axis represents normalized values relative to the values of untreated control cells. F) Secreted MUC5AC from differentiated control (black circles), KChIP3-GFP (red circles) and KChIP3-MUT (green circles) cells that were incubated for 30 minutes at 37ºC in the in the presence (1.2 mM CaCl_2_) or absence (0.5 mM EGTA) of extracellular Ca^2+^. Data were normalized to intracellular actin levels. The y-axis represents normalized values relative to the values of untreated control cells. G) Secreted MUC5AC from differentiated control (black circles), KChIP3-GFP (red circles) and KChIP3-MUT (green circles) cells after 30 minutes incubation at 37ºC with vehicle or 10 µM dandrolene (DAND) in the presence of extracellular Ca^2+^. Data were normalized to intracellular actin levels. The y-axis represents normalized values relative to the values of untreated control cells. Abbreviations: EGTA: Buffer with 0.5 mM EGTA, DAND: 10 µM Dandrolene treatment. * p< 0.05, ** p<0.01, n.s.: not statistically significant.”

“*Figure 3. KChIP3 localized to a pool of MUC5AC granules*

Differentiated KChIP3-GFP (A) and KChIP3-MUT (B) cells were processed for cytosolic washout after 30 minutes at 37ºC of treatment with 1.2 mM CaCl_2_, 0.5 mM EGTA or 10 µM dandrolene. After fixation and permeabilization, samples were analyzed by immunofluorescence microscopy with an anti-GFP (KChIP3, red), anti-MUC5AC antibody (MUC5AC, green) and DAPI (blue). Images represent a single plane (xy), a zoom of the area within the white square and an orthogonal view of each channel (xz). Scale bar = 5 µM. White arrows point to the colocalization between KChIP3 and MUC5AC. Abbreviations: EGTA: Buffer with 0.5 mM EGTA, DAND: 10 µM dandrolene treatment.”

“*Figure 4. KChIP3^-/-^ mice show increased mucus layer thickness*

A, C) Representative distal colons of WT (left panel) and KChIP3^-/-^ (right panel) mice stained with PAS (A) or PAS-AB (C) at different magnification (2.5X, 10X and 40X). B, D) Quantification of the mucus layer thickness in the distal colon stained with PAS (B) or PAS-AB (D) of WT (black dots) and KChIP3^-/-^ (blue dots) mice. Average values ± SEM are plotted as scatter plot with bar graph. The y-axis represents the thickness of the mucus layer in µm. Abbreviations: +/+: WT mice, -/-: KChIP3^-/-^ mice, ML: Mucus layer, GC: Goblet cell. * p<0.05.”

**“***Figure 5. Model of KChIP3 function*

Post biogenesis at the Golgi, mucin granules undergo maturation and are placed on path to fuse with the plasma membrane and release their contents. However, the mature granules recruit KChIP3 (unbound to Ca^2+^) and this event stalls their fusion to the plasma membrane. KChIP3 recruitment thus acts as a brake to control the number of granules that can fuse with the plasma membrane. Based on our findings we describe two situations: A) In absence of intracellular Ca^2+^ oscillations, KChIP3 is bound to the granules preventing their secretion, and B) Upon a rise in intracellular Ca^2+^ by RYR-induced Ca^2+^ oscillations, the KChIP3 on the granules binds Ca^2+^ and subsequently detaches thereby triggering their SNARE dependent fusion to the plasma membrane. The cells have therefore evolved a mechanism by imposing the recruitment of empty KChIP3 to mucin granules to stall their progression leading to fusion with plasma membrane. Abbreviations: Cyt.: cytosol, Ext.: extracellular medium.”

“*Figure 1—figure supplement 1. KChIP expression levels in HT29-18N2 stable cell lines*

A) KChIP1, KChIP2, KChIP3 and KChIP4 RNA levels normalized to values of GAPDH from control and KChIP3-KD cells. mRNA levels of each gene are represented as relative value compared to control cells. Results are average values ± SEM (N ≥ 3). B) Cell lysates from control, KChIP3-GFP and KChIP3-MUT HT29-18N2 differentiated cells were analysed by western blot with an anti-KChIP3 and an anti-GFP antibody to test expression levels. Actin was used as a loading control. C) RNA levels of KChIP1, KChIP2, KChIP3 and KChIP4 (normalized to values of the HPRT1) from control and KChIP3-GFP cells. mRNA levels of each gene are represented as relative value compared to control cells. Results are average values ± SEM (N ≥ 3). Abbreviations: CTRL: control cells. **p<0.01.”

"*Figure 1—figure supplement 2. KChIP3 levels regulate MUC2 secretion*

A) Secreted MUC2 from differentiated control (black circles) and KChIP3 stable knockdown cells (KChIP3-KD) (blue squares) after 30 minutes incubation at 37ºC in normal buffer (1.2 mM CaCl_2_). Data were normalized to actin levels. The y-axis represents normalized values relative to the values of untreated control cells. Average values +- SEM are plotted as scatter plot with bar graph (N=3). B) Secreted MUC2 from differentiated control (black circles), KChIP3-GFP overexpressing cells (KChIP3-GFP) (red circles) and KChIP3-MUT overexpressing cells (KChIP3-MUT) (green circles) after 30 minutes incubation at 37ºC in normal buffer (1.2 mM CaCl_2_). Data were normalized to actin levels. The y-axis represents normalized values relative to the values of untreated control cells. Average values ± SEM are plotted as scatter plot with bar graph (N=3). C) MUC5AC internal protein levels in differentiated control and KChIP3-KD cells normalized to the actin levels. Average values ± SEM are plotted as scatter plot with bar graph (N=3). D) MUC5AC internal levels in differentiated control, KChIP3-GFP and KChIP3-MUT cells normalized to actin levels. Average values ± SEM are plotted as scatter plot with bar graph (N=3). *p<0.05, **p<0.01.”

“*Figure 2—figure supplement 1. KChIP3 does not control Ca^2+^ levels*

A) Time course of Ca^2+^ responses (normalized FURA-2AM ratio) (in grey individual cells, in red average trace) obtained in differentiated HT29-18N2 cells in a buffer containing 0.5 mM EGTA to trigger high levels of Ca^2+^ oscillations. After 5 minutes of recording, cells were treated with vehicle (DMSO) (left) or 10 µM dandrolene (right). At minute 15, cells were stimulated with 100 µM ATP. B) Percentage of cells oscillating in each condition presented in (A) during 10 minutes after starting the treatment. Results are average values ± SEM (N>3). C) Time course of mean basal intracellular Ca^2+^ levels (normalized FURA-2AM ratio) in KChIP3-KD and control cells. D) Time course of mean basal intracellular Ca^2+^ levels (normalized FURA-2AM ratio) in KChIP3-GFP and control cells. E) Time course of mean Ca^2+^ responses (normalized FURA-2AM ratio) obtained in differentiated control (black) and KChIP3-KD (white) cells treated with 100 µM ATP (left panel). Average AUC (Area Under the Curve) from the traces are showed as a bar graph. F) Time course of mean Ca^2+^ responses (normalized FURA-2AM ratio) obtained in differentiated control (black) and KChIP3-GFP (white) cells treated with 100 µM ATP (left panel). Average AUC (Area Under the Curve) from the traces are showed as a bar graph. G) Percentage of cells oscillating in KChIP3-KD, KChIP3-GFP and control cells during 10 minutes in normal buffer (1.2 mM CaCl_2_). Abbreviations: Veh: Vehicle (DMSO), Dand: dandrolene. *p<0.05.”

“*Figure 3—figure supplement 1. KChIP3 apical localization in HT29-18N2 goblet cells*

A) Differentiated KChIP3-GFP and KChIP3-MUT cells were processed for cytosolic washout, fixed and permeabilized for analysis by immunofluorescence microscopy with an anti-KChIP3 (red), anti-GFP antibody (green) and DAPI (blue). Images represent a single plane (xy) and an orthogonal view of each channel (xz). Scale bar = 5 µM. B) Colocalization between KChIP3 and GFP was calculated from immunofluorescence images by Manders’ coefficient using FIJI. Average values ± SEM are plotted as scatter plot with bar graph. The y-axis represents Manders’ coefficient of the fraction of GFP overlapping with KChIP3. C) Differentiated KChIP3-GFP and KChIP3-MUT cells were processed for cytosolic washout at t=0. After fixation and permeabilization, samples were analyzed by immunofluorescence microscopy with an anti-GFP (KChIP3, red), anti-MUC5AC antibody (MUC5AC, green) and DAPI (blue). Images represent a single plane (xy), a zoom of the area within the white square and an orthogonal view of each channel (xz). Scale bar = 5 µM. White arrows point to the colocalization between KChIP3 and MUC5AC. D, F) Colocalization between KChIP3-GFP (D) or KChIP3-MUT (F) (anti-GFP) and MUC5AC at different conditions (t=0’, t=30’ 1.2 mM CaCl_2_, t=30’ 0.5 mM EGTA, t=30’ 10 µM DAND) was calculated from immunofluorescence images by Manders’ coefficient using FIJI. Average values ± SEM are plotted as scatter plot with bar graph. The y-axis represents Manders’ coefficient of the fraction of KChIP3 overlapping with MUC5AC. E, G) Volume of KChIP3-GFP (E) or KChIP3-MUT (G) granules divided by the number of cells per field at different conditions (t=0’, t=30’ 1.2 mM CaCl_2_, t=30’ 0.5 mM EGTA, t=30’ 10 µM DAND). Volume was calculated using 3D objects counter from FIJI software. Abbreviations: EGTA: buffer with 0.5 mM EGTA, DAND: 10 µM Dandrolene treatment. *p<0.05.”

“*Figure 4—figure supplement 1. KChIP3^-/-^ mice show increased mucus layer at the medial colon*

A) KChIP1, KChIP2, KChIP3 and KChIP4 RNA levels were evaluated in the colon of both KChIP3^-/-^ and WT mice by quantitative real-time PCR and normalized to HPRT1 levels. mRNA levels of each gene are represented as relative value compared to WT mice. Results are average values ± SEM (N ≥ 3). B) Representative medial colons of WT (left panel) and KChIP3^-/-^ (right panel) mice stained with PAS or PAS-AB at different magnification (2.5X and 10X). C, D) Quantification of the mucus layer thickness in the distal colon stained with PAS (C) or PAS-AB (D) of WT (black dots) and KChIP3^-/-^ (blue dots) mice. Average values ± SEM are plotted as scatter plot with bar graph. The y-axis represents the thickness of the mucus layer in µm. Abbreviations: WT: WT mice, KO: KChIP3^-/-^ mice.”

3) Regarding calcium oscillations, are the authors certain that it is the oscillations themselves that are responsible for regulating mucin secretion versus mean cytoplasmic calcium concentrations?

We propose that to promote baseline mucin secretion, intracellular calcium needs to reach a certain threshold (which would depend on KChIP3’s Ca^2+^ binding properties). This threshold is reached by Ca^2+^ oscillations generated upon RYR mediated Ca^2+^ release from the ER, which is in close proximity to mucin granules in goblet cells (Tuvim, 2009). So, Ca^2+^ oscillations are likely generated in the proximity of mucin granules and not diffused throughout the cytoplasm. It is also possible that not all oscillations trigger mucin secretion, but only those that exhibit a higher level. Therefore small increases in mean cytoplasmic calcium concentration are likely not sufficient to trigger mucin secretion. Besides, KChIP3-KD cells show increased and KChIP3-GFP cells show decreased baseline mucin secretion despite presenting no differences in mean baseline cytoplasmic calcium concentration (Figure 2—figure supplement 1C and 1D).

4) First sentence of the Abstract "Regulated secretion by specialized goblet cells in agonist-dependent (stimulated) and -independent (baseline) manners.…".If secretion is "regulated", then it should not occur in an agonist-independent manner, and it probably doesn't. The authors are correct when they assert that even baseline secretion is regulated, at least in airway cells, as demonstrated several ways. First, baseline secretion depends on the exocytic sensor of second messengers Munc13-2 (Zhu et al., 2008). Second, it depends on the second messenger calcium (Rossi, Sears and Davis, 2004). Third, it depends on the P2Y2 purinergic receptor (references in Davis and Dickey, 2008). Fourth, it depends on shear stress, probably mediated by ATP released mostly by neighboring ciliated cells during respiration (Zhu et al., 2015; Tarran R, Annu Rev Physiol, 2006). A simple way to frame this is in terms of low and high levels of agonists (or second messengers) rather than agonist-independent and agonist-dependent. Presumably there is a constitutive secretory pathway that exists in parallel to the baseline and stimulated regulated pathways, and this reviewer is aware of some presumptive molecular components of the constitutive pathway in unpublished work. (Oddly, the authors seem to have awareness of these issues because, in their prior work, Mitrovic et al., 2013, the authors correctly noted that "fusion of MUC5AC-containing granules with the plasma membrane requires an external signal."). Please rewrite the Abstract and Introduction with care to make these points more clearly.

Although both baseline and stimulated secretion are regulated processes that depend on secondary messengers, we use the term baseline secretion to describe mucin release in the absence of exogenously applied agonist (as previously defined in Zhu et al., 2015). However, we agree that referring to baseline secretion as agonist-independent event is not completely correct and may be confusing. Therefore, we have changed the first sentence of the Abstract as well as the corresponding points of the Introduction and Discussion as shown below:

Abstract

“Regulated mucin secretion by specialized goblet cells in exogenous agonist-dependent (stimulated) and -independent (baseline) manner is essential for the function and protection of the epithelial lining.”

Introduction

“Mucin granule exocytosis is a Ca^2+^-regulated process that can occur at basal level (Baseline Mucin Secretion, BMS) and by an exogenously applied agonist-dependent release (Stimulated Mucin Secretion, SMS) (Adler et al., 2013; Rossi et al., 2004).”

Discussion

“Specialized goblet cells release heavily glycosylated mucins from stored granules via exogenously applied agonist-dependent and -independent pathways.”

5) In the second to last sentence of the Abstract, the statement "KChIP3 emerges as the long-sought high-affinity Ca^2+^ sensor that regulates baseline mucin secretion" is likely to confuse those in the field because KChIP3 appears to be purely a negative regulator. Syt2 was identified as the low-affinity calcium sensor in stimulated secretion in airway cells, as correctly noted by the authors. However, Syt2 is a positive regulator of stimulated secretion in both airway cells and neurons, because its deletion in airway mucous cells results in a severe reduction of ATP-stimulated mucin release (Tuvim et al., 2009) and its deletion in neurons results in a severe reduction of evoked release in neurons (Pang ZP, J Neurosci, 2006). Besides its positive activity, Syt2 also has a negative regulatory activity because its deletion in neurons results in increased asynchronous and spontaneous synaptic vesicle release, but whether it has a negative activity in mucous cells is not known. In any case, what has long been sought is the high-affinity calcium sensor that positively regulates baseline mucin secretion. While Cantero-Recasens et al. have apparently discovered an interesting regulatory pathway, they need to explain these differences with care as not to confuse the reader.

We have changed the sentence in the Abstract to: " KChIP3 therefore emerges as the high-affinity Ca^2+^ sensor that negatively regulates baseline mucin secretion." Moreover, we have included a new paragraph in the Discussion to clarify the role of KChIP3 as a negative regulator of baseline mucin secretion.

Discussion

“We propose that KChIP3 is the high-affinity calcium sensor to control baseline mucin secretion. KChIP3 is, however, a negative regulator of this process. Syt2, on the other hand, is the low-affinity calcium sensor that acts as a positive regulator of stimulated mucin secretion, since its deletion severely blocks ATP-stimulated mucin release in airways cells (Tuvim et al., 2009). Why is baseline mucin secretion controlled by a negative regulator? One possibility is that granules for baseline secretion are already docked to the membrane and ready to release their contents. The only means to prevent their fusion therefore is to employ a brake –a negative regulator– such as KChIP3. Another possibility that we cannot exclude is the existence of a positive regulator that competes with KChIP3 to regulate baseline mucin secretion.”

6) In the Discussion, it is stated, "in a nutshell, all that is known (about baseline mucin secretion) is that it is independent of Syt2 and is sensitive to mechanical stress." However, we also know that SNAP23 mediates both baseline and stimulated secretion (Ren et al., 2015), and that Munc13-2 regulates both processes (Zhu et al., 2008). Please cite these papers.

We have included the references suggested in the text.

Discussion

“In a nutshell, all that is known is that in human bronchial epithelial cells and mouse trachea, baseline mucin granule secretion is SNAP23 and Munc13-2 dependent (that also regulate stimulated secretion) (Ren et al., 2015; Zhu et al., 2008), but independent of Syt2, which functions during agonist-induced mucin secretion, and it is sensitive to mechanical stress (Zhu et al., 2015). But how is baseline mucin secretion controlled to prevent mucin hyper- or hyposecretion thereby avoiding pathological quantities of secreted mucins?”

7) The authors go back and forth between airway and intestinal goblet cells as if they have identical mechanisms mediating and regulating secretion when in fact they don't. To begin, airway secretion, both baseline and stimulated, depends upon shear forces (airflow in vivo, liquid flow in vitro) mediated by ATP release from neighboring cells, whereas in the intestine, there is an adherent layer of MUC2. Another example is that in the intestine, sentinel goblet cells at the entrance to crypts respond to TLR signaling with massive secretion (Birchenough GM, Science, 2016), whereas no such phenomenon has been detected in the airway. Therefore, the authors should better clarify what is known in the airway, what is known in the intestine, and what, exactly, they are modeling. There is space to do this.

We have included a new paragraph on the Discussion.

Discussion

“In addition, we propose that the basic mechanism of KChIP3 regulating baseline secretion is shared between different tissues, although there are several differences regarding mucin secretion from colonic and airway cells. It is known that both stimulated and baseline secretion in the airways depend on shear stress (airflow and mucus movements promoted by ciliated cells) and are apparently independent of extracellular Ca^2+^ (Davis and Dickey, 2008; Tuvim et al., 2009). In the intestine, there are two layers of mucus organized by MUC2, an outer loose layer (where commensal bacteria live) and an inner adherent layer (which is free of pathogens) (Johansson et al., 2011), and extracellular Ca^2+^ is required for stimulated secretion (Mitrovic et al., 2013). Additionally, the sentinel goblet cells in the colon respond to TLR (Toll-like receptor) ligands and release massive amounts of mucin (Birchenough GM, Science, 2016). Nevertheless, mucin secretion in both tissues is dependent on intracellular Ca^2+^. This necessitates a low affinity Ca^2+^ sensor for stimulated secretion and a high affinity Ca^2+^ sensor for baseline secretion. Thus, KChIP3 is likely to play an important role not only in the physiology and pathophysiology of colon but also in the airways and mucin related pathologies such as asthma or COPD.”

8) Minor issues: Subsection “KChIP3^-/-^ mice show increased colonic mucus layer under basal conditions”, second paragraph, it is stated that an increase in the mucus layer was detected in the medial colon of four out of five knockout mice; was the fifth mouse excluded from the data analysis? In the next paragraph, the comment in Results that "our data showing an increased mucus layer in KO mice fit nicely with data obtained in vitro" is more appropriate for Discussion.

We did not exclude the fifth mouse from the data analysis, since it was included in the analysis of distal colon and there was no objective reason to do so. By removing this mouse the differences in the medial colon between WT and KO mice become statistically significant for both PAS and PAS-AB staining. We have removed the sentence from Results.

9) In the Introduction, it is confusing to state, "our genome-wide screen identified new proteins that control extracellular calcium entry into cells.… also identified was a high-affinity calcium-binding protein, KChIP3". This makes it sound as if KChIP3 regulates calcium entry, when it is subsequently demonstrated that it does not. It should also be pointed out that in airway mucous cells over short time frames, extracellular calcium does not contribute to the level of cytoplasmic calcium (referenced in Tuvim et al., 2009, and Davis and Dickey, 2008).

We have changed it: “Our genome-wide screen identified new proteins that regulate mucin secretion, such as TRPM5, a Na^+^ channel that controls extracellular Ca^2+^ entry into cells (Mitrovic et al., 2013).”

We have stated that extracellular calcium does not contribute to airway mucin secretion in a new paragraph of the Discussion:

“In addition, we propose that the basic mechanism of KChIP3 regulating baseline secretion is shared between different tissues, although there are several differences regarding mucin secretion from colonic and airway cells. It is known that both stimulated and baseline secretion in the airways depend on shear stress (airflow and mucus movements promoted by ciliated cells) and are apparently independent of extracellular Ca^2+^ (Davis and Dickey, 2008; Tuvim et al., 2009). In the intestine, there are two layers of mucus organized by MUC2, an outer loose layer (where commensal bacteria live) and an inner adherent layer (which is free of pathogens) (Johansson et al., 2011), and extracellular Ca^2+^ is required for stimulated secretion (Mitrovic et al., 2013). Additionally, the sentinel goblet cells in the colon respond to TLR (Toll-like receptor) ligands and release massive amounts of mucin (Birchenough GM, Science, 2016). Nevertheless, mucin secretion in both tissues is dependent on intracellular Ca^2+^. This necessitates a low affinity Ca^2+^ sensor for stimulated secretion and a high affinity Ca^2+^ sensor for baseline secretion. Thus, KChIP3 is likely to play an important role not only in the physiology and pathophysiology of colon but also in the airways and mucin related pathologies such as asthma or COPD.”

10) There is room for improvement in phrasing in nearly every paragraph, with some being important (see below) and some trivial, for example, starting with the Abstract, granules are not released, rather their contents are released, and this could simply be fixed as "allowing for their exocytosis".In the Introduction, they talk about "a Ca^2+^-mediated reaction to a gel-like form". This is likely to be very confusing to readers because the manuscript revolves around cytoplasmic calcium sensing, while the effect of calcium on mucus gel-like properties is an entirely different subject that does not come up elsewhere in the manuscript. It is true that calcium binds mucins within secretory granules, and then is exchanged for sodium in the extracellular space after mucin secretion, but it would be simplest to just drop this fact from the Introduction.In the third paragraph of the Introduction, it would be helpful to add a few words "identified new proteins, including TRPM5", to orient the reader to the prior work. “entry into the cells” is better as "entry into cells", and “vicinity of exocytic machinery” as "vicinity of the exocytic machinery".The choice of KChIP3-WT as the designation of the transgenic overexpressing line is confusing, because WT can refer either to WT sequence or WT level of expression; it would be better to designate it as OE for overexpressing or GFP since that is now inserted.Subsection “KChIP3 is required for baseline mucin secretion in colonic goblet cells”, second paragraph, it would be helpful if the authors provided a short description of circumstances when MUC5AC is expressed in colonic epithelium, such as parasite infection or ulcerative colitis, as they did in Mitrovic.

Thanks, we have corrected them (Abstract and Introduction), we refer now KChIP3-WT cell line as KChIP3-GFP, and now we included the following:

“MUC5AC secretion by colonic cancer cells is a good model system to study the mucin secretory pathway and, although it is expressed at low levels in the gastrointestinal and upregulated in pathological conditions such as ulcerative colitis or parasitic infection (Forgue‐Lafitte et al., 2007; Hasnain et al., 2011), colonic goblet cells ordinarily secrete MUC2. This raises the obvious issue: is KChIP3 involved in baseline MUC2 secretion?”

The apparent rigor of statistical analysis is low.

The description of the statistical analysis is included in the Materials and methods and assessed to the best of our abilities.

Reviewer #3:

Cantero-Recasens and colleagues presented an interesting study on the impact of KChIP3 on mucin secretion. They showed that knocking down of KChIP3 lead to mucin hypersecretion, indicating an inhibitory role of KChIP3 in baseline secretion. In support of this notion, the authors showed that overexpression of KChIP3 WT or Ca^2+^ Mut inhibits baseline mucin secretion. Furthermore, the authors found that both KChIP3 WT and Ca^2+^ Mut associate with MUC5AC granules. Increases in Ca^2+^-oscillations by EGTA treatments significantly reduced the granule localization of KChIP3 WT. In contrast, KChIP3 Ca^2+^ Mut showed no changes upon EGTA treatments. These findings have led the authors to conclude that KChIP3 is likely to be the high-affinity Ca^2+^ sensor for baseline mucin secretion, and they further proposed that KChIP3 marks mature and primed mucin granules, and functions as a Ca^2+^-dependent brake for baseline mucin secretion. In general, the experiments in this study were carefully designed and well executed. The inhibitory activity of KChIP3 on baseline mucin secretion is nicely shown. A potential weakness of this study is the link between Ca^2+^ and the dissociation of KChIP3 from granules.KChIP3 granule localization is interesting, and this appears to be the main evidence for the proposed mechanism underlying KChIP3 inhibition. However, it is unclear to this reviewer what is the mechanistic nature behind the reduction of KChIP3 localization upon the EGTA treatment. For example, KChIP3 is a multifunctional protein that binds proteins and DNA, and it regulates gene expression in a Ca^2+^-dependent way. Can the authors exclude the possibility that the changes of KChIP3-granule localization are due to alterations of protein/membrane components of granules on a longer timescale? Also, have the authors tried to block granule exocytosis by knocking down SNAREs (e.g., VAMP8)? Does an impairment of granule exocytosis inhibit KChIP3 dissociation? At this stage, further evidence to support the model in Figure 5 would be helpful.

Although we cannot completely rule out the possibility that KChIP3 localization is affected by alterations in the chemical composition of the granules, the fact that KChIP3-MUT (that cannot bind Ca^2+^) has a similar localization which is not affected by Ca^2+^ levels points to a fast effect of Ca^2+^ treatments on KChIP3 localization. Also the time of our experiments (30 min) makes it unlikely that the effect of Ca^2+^ treatment on KChIP3 localization is due to alterations in membrane compositions.

We have not tested the effect of SNAREs KD (like VAMP8, SNAP23 or the exocytic regulatory protein Munc13-2 that are known to control baseline mucin secretion) in KChIP3 cell lines. Interestingly, all these SNAREs identified so far control both baseline and stimulated mucin secretion in the airways. However, the specific Syntaxin for baseline or stimulated mucin secretion remains unknown. Our immediate goal is to identify the syntaxin protein involved in baseline secretion and then test whether its depletion triggers accumulation of granules in KChIP3 lacking cells. We also believe a better understanding of how KChIP3 is recruited to the granules by identification of its binding partners, will help address this issue more convincingly. But these studies are beyond the scope of this paper.

In Figure 3, the authors presented data on KChIP3 staining using permeabilized and washed cells. While it is clear that some KChIP3 proteins are co-localized with granules, the degree of KChIP3 granular/cytosolic localization is unclear. What percentage of total KChIP3 is on granules? Without washing, could the authors observe KChIP3 WT and Ca^2+^ Mut on docked granules (e.g., by TIRF)? Does the EGTA treatment change KChIP3 localization in intact cells? It would be helpful to know whether the EGTA treatment alters KChIP3 distribution by triggering complete dissociation or by changing the tightness of KChIP3-granule interactions.

Our first experiments using non-cytosolic washed KChIP3-GFP cells showed cytosolic KChIP3 localization with some brighter punctae of KChIP3, so we decided to check its localization by first removing all non-membrane bound cytoplasmic protein.

We hypothesize that binding of Ca^2+^ to KChIP3 (provided by calcium oscillations that reach a certain threshold) changes its conformation and thereby affects its interaction to mucin granules. For example, it is known that binding of Ca^2+^ to KChIP3 releases it from DRE sites (where it acts as a transcriptional repressor) in neurons (Carrión, 1999). However, whether this possible conformational change after binding Ca^2+^ triggers dissociation from the granules, alters the binding affinity of KChIP3 to granules, increases the probability of its release, or produces a change in its binding partners are beyond the scope of this paper.